# Disruption of tRNA threonylation triggers RIG-I mediated anti-tumour immune response

Cléa Dziagwa[1], Christian Seca[1,13], Coralie Capron [2,13], Chloe Maurizy[3], Ning An[1], Denis Heusdens[4], Timothy Budden[5,6,7], Lorena Martin-Morales [2], Miguel Susaeta Ruiz [1], Elodie Renaude[4], Najla El-Hachem [1], Raphael Vanleyssem[1], Marine Leclercq [1], Arnaud Blomme [4], Alain Chariot [3,8], Jochen Utikal[9,10,11], Amaya Viros [5,6], Francesca Rapino[2,8], Sylvain Delaunay[1,12,14] & Pierre Close [1,8,14] ✉

Tumour-induced mechanisms of immune evasion hinder immune response to cancer, particularly in melanoma. mRNA translation, by ensuring accurate protein synthesis, regulates cancer phenotypes and immune response, but the underlying mechanisms remain unclear. Here, we reveal how O-sialoglycoprotein endopeptidase (OSGEP), catalysing the tRNA modification N[6]-threonylcarbamoyladenosine (t[6]A), drives protein homeostasis in cancer cells to maintain T-cell exclusion and prevent anti-tumour immune response. t[6]A-deficient melanoma cells disrupt efficient cytoplasmic translation of ANN codons (trinucleotides with A in the first position and N = any nucleotide), causing specific protein aggregation and the formation of integrated stress response-dependent stress granules. We discovered that OSGEP loss triggers melanoma regression by relocating RIG-I to stress granules, leading to its pathway activation. As a result, T-cells are recruited to the tumour site and orchestrate an anti-tumour immune response. Finally, an OSGEP-driven gene signature in melanoma patients is associated with T-cell infiltration and improved overall survival. Together, our findings position t[6]A tRNA modification as a promising therapeutic target for melanoma treatment.

Immune response in cancer is a multifaceted and dynamic process that plays a critical role in the development, progression, and treatment of various malignancies[1]. Immune surveillance allows for the recognition and eradication of tumour cells. However, cancers have evolved sophisticated strategies to evade immune detection, promoting tumour development and posing significant barriers to the efficacy of immunotherapies[2–4]. Specifically in melanoma, these immune evasion strategies are particularly pronounced due to a complex immuno-suppressive tumour microenvironment (TME)[5,6]. T-cells are fundamental components of adaptive immunity, playing key roles in antigen recognition, coordination of immune responses, and modulation of

immune tolerance across diverse physiological contexts. Their ability to recognize and eliminate cancer cells through cytotoxic activity positions them as critical mediators of tumorigenesis. The effectiveness of T cell-mediated responses is often hampered by the TME, which can inhibit T-cell recruitment and activation through various mechanisms (e.g. expression of inhibitory ligands or repression of cytokine secretion)[7,8].

Proteostasis (i.e. protein homoeostasis) is crucial in modulating immune response in cancer[8–11]. Efficient mRNA translation ensures the production of correctly folded and functional proteins to adapt to changes in the TME[12]. For instance, MYC induces immune evasion in

liver cancer by translationally upregulating programmed death-ligand 1 (PD-L1)[13]. However, disrupted translation can lead to the generation of misfolded proteins, which accumulate in aggregates[14], ultimately affecting the tumour immunogenicity and the course of the disease[15]. Yet, the precise mechanisms through which tumour cells maintain proteostasis to evade immune response remain largely unknown.

Deposition of post-transcriptional modifications on tRNAs is critical for their stability and function, with modifications in the anticodon region being particularly important for the accuracy of protein synthesis[16–18]. The landscape of these modifications is frequently altered in cancer, as a mechanism to adapt to both global and specific changes in mRNA translation[19]. The N6-threonylcarbamoylation of adenosine at position 37 (t6A) is a universally conserved modification necessary for the efficient decoding of ANN-accepting tRNAs[20–23]. The t6A biosynthesis pathway involves the KEOPS protein complex (Kinase, Endopeptidase and Other Proteins of Small size), with OSGEP (O-sialoglycoprotein endopeptidase) serving as the catalytic subunit responsible for the transfer of the threonylcarbamoyl moiety[24]. Loss-of-function mutations in genes encoding KEOPS complex subunits, including OSGEP, have been linked to Galloway–Mowat syndrome (GAMOS), a rare neuro-renal disorder characterised by a combination of steroid-resistant nephrotic syndrome and microcephaly with neurological impairments[25–27]. The functions of the t6A tRNA modifications in cancer and immunity are currently largely unknown.

RIG-I (Retinoic acid-inducible gene I) is a crucial sensor of innate immune system, primarily known for its role in detecting viral RNA and initiating antiviral responses in infected cells[28–31]. Recent studies have demonstrated that RIG-I activation triggers a potent anti-tumour immune response[32,33]. Upon recognition of aberrant RNAs within tumour cells, RIG-I initiates a signalling cascade that leads to the production of type I interferons, pro-inflammatory cytokines, and chemokines. This immune response enhances the recruitment and activation of immune effector cells at the tumour site[34,35]. Consequently, RIG-I serves as a critical immune sensor for anti-tumour immunity and represents a promising target for the development of cancer immunotherapies.

Here, we reveal that OSGEP and t6A tRNA modification are essential for the maintenance of proteostasis and thereby regulate immune reaction in melanoma. We show that OSGEP depletion in cancer cells promotes T-cells recruitment and activation at the tumour site and effectively prevents tumour growth. Mechanistically, low levels of t6A compromise the efficient decoding of ANN codons during translation. OSGEP knockdown leads to the generation of protein aggregates and initiates the activation of the Integrated Stress Response (ISR) effector Heme-Regulated Inhibitor Kinase (HRI), driving stress granule formation. RIG-I is activated within these stress granules and orchestrates a robust T cell-dependent anti-tumour response. Together, our study establishes that the t6A tRNA modification plays a critical role in preventing RIG-I activation and mitigating the anti-tumoral immune response in melanoma.

## Results

### A targeted proteostasis screen identifies the essential role of OSGEP in tumour immune response
RNA modifications located at the anticodon loop of cytoplasmic tRNAs allow for the optimal decoding of mRNAs during translation and promote the maintenance of protein homoeostasis[16,17,36,37]. To identify anticodon tRNA modifying enzymes (tRME) required for proteostasis, we generated an esiRNA (i.e. *endoribonuclease-prepared siRNA, a pool of siRNAs targeting the same mRNA sequence*) screen library targeting the currently known tRMEs responsible for chemical deposition on the anticodon loop of cytoplasmic tRNAs ($n = 45$; Fig. 1a)[38]. 48h after transfection, the generation of protein aggregates was measured by flow cytometry as a readout of mistranslation in two patient-derived melanoma cell lines (i.e. MM117, MM011). The esiRNA targeting OSGEP led to the highest generation of aggregates in both patient-derived lines (Fig. 1b, and Supplementary Fig. 1a). Using specific shRNAs, we validated that OSGEP depletion induced an increase of protein aggregation propensity in both MM117 and MM011 patient-derived melanoma lines, as well as in two additional murine melanoma cell lines (i.e. M1014, B16F10; Fig. 1c, d). This enzyme is the catalytic subunit of the KEOPS complex, responsible for the modification t6A in position 37 of cytoplasmic tRNA^NNU[24–26,39] (Supplementary Fig. 1a).

B16F10 cells expressing or lacking OSGEP were then subjected to transcriptomics analyses. Interestingly, OSGEP depletion led to the activation of immune pathways, such as cytokine production, T cell-mediated immunity and immune system processes (Fig. 1e, Supplementary Fig. 1b–d, and Supplementary Datasets 1 and 2). We hypothesized that the t6A tRNA modification is necessary for tumour immune regulation by ensuring efficient mRNA translation and maintenance of proteostasis. To test this, we depleted OSGEP in B16F10 cells and implanted these cells into either immune-deficient (NOD-SCID) or immune-competent (syngeneic C57BL/6) mice. Its knockdown significantly reduced B16F10 and M1014 tumour size at day 14 in immune-competent mice but did not impact B16F10 tumour development in immune-deficient mice (Fig. 1f, g, and Supplementary Fig. 2a–d). OSGEP depletion greatly delayed melanoma development and extended survival of melanoma-bearing immune-competent individuals (Supplementary Fig. 2a–d). Further histological staining of tumours showed a decrease in proliferation (Ki67) and an increase of apoptosis (cleaved Caspase-3) in OSGEP-depleted tumours in immune-competent mice, while no difference was observed in immune-deficient mice (Fig. 1h–I, and Supplementary Fig. 2g,h). Finally, depletion of L antigen Family Member 3 (LAGE3), another member of the t6A KEOPS complex, led to a reduction of B16F10 tumour size in immuno-competent mice, without affecting tumour development in immuno-deficient mice (Supplementary Fig. 2e–f). Together, our data demonstrated that depletion of OSGEP, as part of the KEOPS complex, compromises cellular proteostasis and triggers an immune-mediated anti-tumoral response in melanoma.

### OSGEP prevents T-cell recruitment and activation in the primary tumour
To decipher how OSGEP expression controls the tumour-associated immune composition, we collected immune cells by flow cytometry from control and OSGEP-deficient tumours of immunocompetent recipients at 14 days post transplantation. In line with prior reports, B16F10 tumours were poorly infiltrated and 'immune cold' (Fig. 2a, b, and Supplementary Figs. 3a and 4a)[40,41]. However, tumours lacking OSGEP displayed robust infiltration of T cells, without affecting other immune cell types, such as B cells, macrophages or neutrophils (Fig. 2a, and Supplementary Figs. 3b–e and 4b–f). Consistently, immunofluorescence CD3 staining in tumours confirmed abundant T cell infiltration in OSGEP-deficient tumours (Fig. 2b). Further flow cytometry analysis also revealed that both CD4+ and CD8+ T cells were enriched in tumours lacking OSGEP and had significantly higher capacity to produce IFNγ and TNFα (features of activated CD4+ and CD8+ effector T cells) compared with control tumours (Fig. 2c, d). Finally, to confirm the functional importance of T cells in the anti-tumour response triggered in OSGEP-defective cells, we depleted CD8+ and CD4+ T cells (i.p. injection of monoclonal anti-CD8/anti-CD4 antibodies) in C57BL/6 mice before tumour inoculation (Fig. 2e, and Supplementary Fig. 5a). T cells depletion abolished the tumour growth defect seen upon OSGEP knockdown (Fig. 2f, g). Of note, NK cell depletion could not rescue the growth of OSGEP-deficient tumours (Supplementary Fig. 5b–d). Collectively, these data indicate that effector CD4+ and CD8+ T cells actuate the rejection of tumours lacking OSGEP.

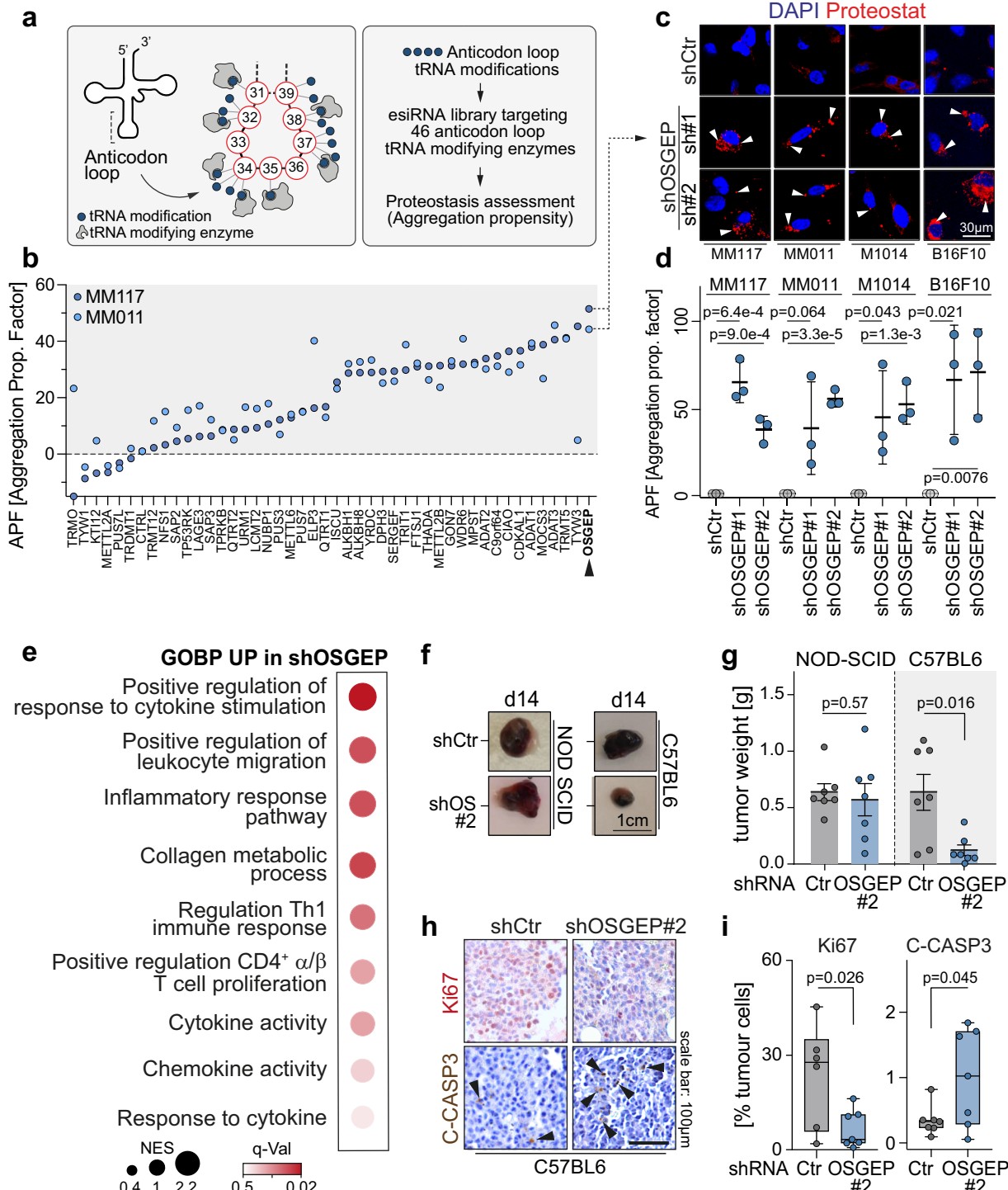

**Fig. 1 | OSGEP is required to prevent immune response in melanoma.**
**a** Schematic overview of the proteostasis screen targeting anticodon-loop tRNA modifications. **b** Average aggregation propensity in MM117 and MM011 patient-derived melanoma cell lines, subjected to esiRNA non target, or targeting one of 45 tRNA modifying enzymes (esiRNA: *endoribonuclease-prepared siRNA, a pool of siRNAs targeting the same mRNA sequence).* Immunofluorescence (**c**) and quantification (**d**) of aggregation propensity factor in MM117, MM011, B16F10 and M1014 cells infected with shCtr or shOSGEP stained for proteostat and DAPI as a nuclear counterstain (*n* = 3 independent experiments). **e** Gene set enrichment analysis (GSEA) from RNA-seq of the shCtr versus shOSGEP B16F10 cells (NES: normalized enrichment score; GOBP: gene ontology biological process). **f, g** Representative

B16F10 tumours and quantification of tumour weight, 14 days after orthotopic subcutaneous transplantation in immunodeficient NOD SCID or immunocompetent C57BL/6 mice (*n* = 7 mice per condition). Tumours derived from B16F10 cells were infected with control (shCtr) or OSGEP (shOSGEP) shRNAs. Representative immunostainings (**h**) and quantification (**i**) of KI67 and Cleaved-CASPASE-3 (C-CASP3) staining in B16F10-derived tumours (day 14) infected with shRNAs Ctr or targeting OSGEP (*n* = 7 tumours per condition). Shown is mean +/− SD (**d, g**). Box plots in (**i**) show minimum value, first quartile, median, third quartile and maximum value. Unpaired two-tailed *t*-test (**d, i**). Paired two tailed *t*-test (**g**). Exact *p*-values are indicated. Source data are provided as a Source Data file.

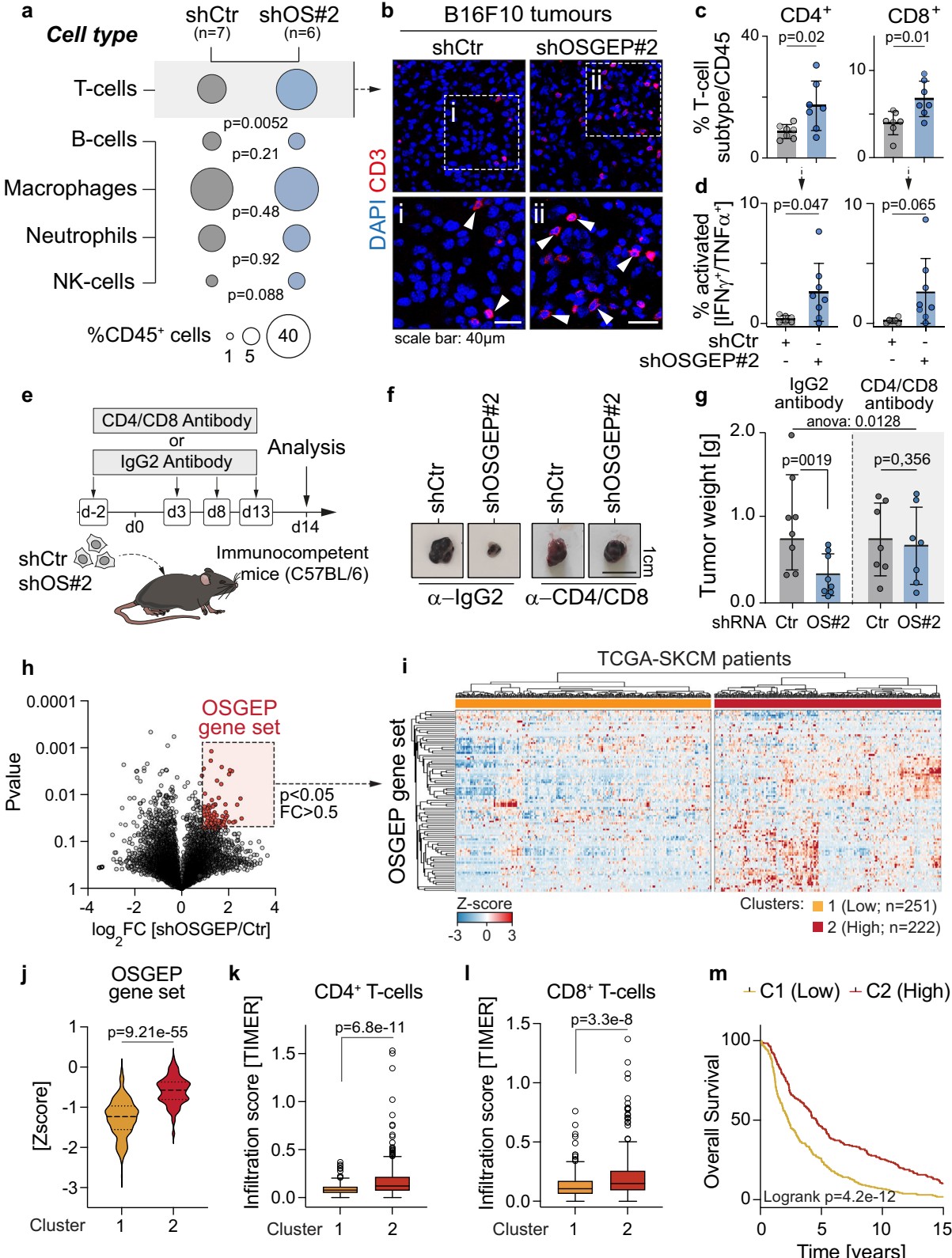

To define the clinical features of OSGEP-driven transcriptional profiles, we collected the upregulated by the depletion of OSGEP in B16F10 cells (i.e. *OSGEP* gene set, log2FC > 0.5 and *p*Value > 0.05; Fig. 2h; Supplementary Dataset 1). We performed an unsupervised clustering analysis which grouped melanoma patients from *The Cancer Genome Atlas* database (TCGA; *n* = 473) into two clusters, according to the *OSGEP* gene set (Fig. 2i, j, and Supplementary

Fig. 6a, b). In line with our observations that OSGEP deficiency in melanoma tumours triggers an immune phenotype mainly resolved by CD3+ T cells (Fig. 2f, g), the infiltration of CD4+ and CD8+ T-cells progressively increased with our OSGEP gene set levels (Fig. 2k, l). Additionally, the overall survival of SKCM patients correlated with the expression of the *OSGEP* gene set (Fig. 2m, and Supplementary Fig. 5e). Notably, OSGEP protein levels were markedly increased in

**Fig. 2 | OSGEP depletion in tumours induces recruitment and activation of T-cells. a** Bubble plot of the percentage of infiltrating T cells, B cells, macrophages, neutrophils and NK cell subpopulations normalised to CD45$^+$ cells (average of $n = 7$ shCtr and $n = 6$ shOSGEP tumours). **b** Immunofluorescence of B16F10-derived tumours (day 14) infected with shCtr (left panels) or shOSGEP (right panels) stained for CD3 and DAPI as a nuclear counterstain. Representative staining of 3 tumours per condition. **c, d** Quantification of the percentage CD4$^+$ (left) or CD8$^+$ (right) T cells, and of activated CD4$^+$ (left) or CD8$^+$ (right) T cells IFN$\gamma^+$/TNF$\alpha^+$ in B16F10 tumours, 14 days after transplantation in immunocompetent mice (shCtr $n = 6$; shOSGEP $n = 8$ mice). **e** Scheme of experimental procedure for the depletion of CD4 and CD8 cells in immunocompetent mice using control (IgG) or antibodies targeting CD4/CD8, starting 2 days before subcutaneous injection of melanoma cells. **f, g** Representative B16F10 tumours and quantification of tumour weight, 14 days after orthotopic subcutaneous transplantation into C57BL/6 mice treated with IgG2 ($n = 8$ mice per condition) or CD4/CD8 ($n = 7$ mice per condition) antibodies.

Tumours derived from B16F10 cells were infected with control (shCtr) or OSGEP (shOSGEP) shRNAs. **h** Volcano plot of the expression of transcripts in shCtr versus shOSGEP B16F10 cells and generation the OSGEP gene set signature consisting of genes with Log2(shOSGEP/shCtr)>0.5 and $p$value < 0.05 and matched with the TCGA dataset ($n = 94$ genes). **i** Heatmap of unsupervised clustering identified 2 clusters of patients with SKCM (TGCA) according to the top 100 upregulated genes from the OSGEP gene set signature (cluster 1, $n = 251$; cluster 2, $n = 222$). Expression of the OSGEP gene set signature (**j**), level of infiltration of CD4$^+$ (**k**) and CD8$^+$ T cells (**l**) in SKCM-TCGA tumours in the clusters identified in **i. m** Survival analysis of the SKCM-TCGA patients in the clusters identified in **i**. Shown is mean +/− SD (**c, d, g**). Box plots in (**j–l**) show minimum value, first quartile, median, third quartile and maximum value. Two-way ANOVA (row factor variation; **g**); Unpaired two-tailed $t$-test (**a, c, d, g, h, k, l**). Paired two-tailed $t$-test (**g, j**). Exact $p$-values are indicated. Source data are provided as a Source Data file.

melanoma lesions compared to matched adjacent skin in human melanoma biopsies (Supplementary Fig. 5f, g).

As an orthogonal confirmation of our observation, we conducted a spatial transcriptomic analysis of melanoma samples from two patients (Supplementary Fig. 6c–f). Our analysis revealed that the level of T-cell infiltration, as assessed by the T-cell signature score, was significantly elevated in Patient_01, which was classified as immunologically 'hot'. Additionally, this 'hot' tumour also exhibited high levels of the OSGEP gene set (Supplementary Fig. 6g, h). We further examined the heterogeneity in OSGEP gene set expression within the tumours and correlated its spatial expression with the proximity of T-cells. Notably, in Patient_01, which displayed high levels of T-cell infiltration, we observed a positive correlation between the expression of the OSGEP gene set and the proximity to T-cells. In contrast, this correlation was absent in Patient_02, which displayed low levels of T-cell infiltration (Supplementary Fig. 6i–k).

In summary, the *OSGEP* signature is prognostic for T-cell tumour infiltration and overall survival in SKCM (skin cutaneous melanoma) patients. Thus, we identified a transcriptome gene signature predicting tumour immune reaction and survival in melanoma.

## OSGEP supresses RIG-I signalling pathway

Changes in the TME by, for instance, the expression and secretion of cytokines and chemokines by tumour cells have been shown to modulate specific Tcell recruitment[42,43]. To assess whether knockdown of OSGEP in melanoma cells alone was sufficient to induce migration of T cells, we performed ex-vivo co-culture experiments. CD3$^+$ T cells were isolated from the spleens of C57BL/6 mice and seeded in Transwell®, with the bottom chamber containing control or OSGEP-deficient B16F10 cells (Fig. 3a, and Supplementary Fig. 7a). We observed that knockdown of OSGEP strongly enhanced migration of naïve T cells ex-vivo, suggesting the loss of OSGEP in melanoma cells triggers the release of chemoattractants (Fig. 3b). To investigate this further, we profiled the cytokinome of the medium of B16F10 cells depleted for OSGEP. Overall, the cytokines secreted, including the ones specifically attracting T cells[44], were increased in the absence of OSGEP (Fig. 3c, and Supplementary Fig. 7b, c). Interestingly, the transcriptome of OSGEP-depleted B16F10 cells also displayed upregulation of pathways related to type I Interferon (IFN-I), defence response to virus and retinoic acid-inducible gene I like receptor (RLR) signalling (Fig. 3d). The latter is a family of intracellular pattern recognition receptors involved in the detection of viral DNA and RNA[45]. This activation results in the production of IFN-I and other pro-inflammatory cytokines, critical for antiviral immune response[28]. To assess the influence of OSGEP loss on the activation of RLR receptors, we measured the protein levels of all members of the RLR family[46]. While Cyclic GMP-AMP synthase (cGAS) remained unchanged, and Melanoma differentiation-associated 5 (MDA-5) and Stimulator of interferon genes protein (STING) levels varied inconsistently among cell lines, retinoic acid-

inducible gene I (RIG-I) was strongly and consistently upregulated by OSGEP or LAGE3 depletion in B16F10 and M1014 melanoma cells, or A475 human melanoma cell (Fig. 3e, and Supplementary Fig. 7d). Moreover, the pathway downstream of RIG-I was activated, with increased phosphorylation levels of TANK-binding kinase 1 (p-TBK1) and nuclear localization of Interferon regulatory factor 3 (IRF3; Fig. 3f–h, and Supplementary Fig. 7e–g)[47]. OSGEP-depleted tumours also upregulated RIG-I expression in vivo in melanoma tumours (Fig. 3i). Finally, RIG-I levels correlated with CD4$^+$ and CD8$^+$ Tcell infiltration in skin cutaneous melanoma tumours (Supplementary Fig. 7h–j). Together, our data reveal that OSGEP loss activates RIG-I signalling pathway.

## t$^6$A tRNA modification allows efficient decoding of ANN codons

Since t$^6$A modification on tRNAs is required for decoding of ANN codons during mRNA translation (Fig. 4a), we asked whether the loss of OSGEP could lead to a disruption in protein synthesis. First, we assessed the level of the 52 most abundant cytoplasmic tRNA modifications by LC-MS. Knockdown of OSGEP in B16F10 cells led to a strong and specific reduction of t$^6$A levels (Fig. 4b, and Supplementary Dataset 3). 2-methylthio-N6-threonylcarbamoyladenosine (ms$^2$t$^6$A) modification, the result of further thiomethylation of t$^6$A sites by Cdk5 regulatory associated protein 1-like 1 (CDKAL1) on tRNA$^{UUU}$[48,49], was also sensitive to OSGEP depletion (Fig. 4b). Of note, the lack of t$^6$A modification did not lead to changes in the tRNA pool in B16F10 cells, showing the deposition of t$^6$A on tRNAs$^{NNU}$ does not affect their stability (Supplementary Fig. 8a–d).

To determine the effect of OSGEP on global mRNA translation, we performed L-homopropargylglycine (HPG) incorporation in B16F10 cells lacking OSGEP[50]. Interestingly, depletion of OSGEP did not affect global protein synthesis, which was in line with previously described studies[21,51] (Fig. 4c). To validate this finding, we assessed the polysome profile of control and OSGEP-depleted B16F10 cells. Here, again, OSGEP depletion did not impact the polysomal fraction (Supplementary Fig. 9a).

During translation, only three codons engage with tRNAs at the ribosome's E-site, P-site, and A-site. Cognate aminoacyl-tRNAs enter the ribosome at the A-site, representing the site of specific codon-anticodon pairing, markedly affected by the presence of modifications at the tRNA anticodon loop. To assess the effect of OSGEP on codon decoding, we mapped the ribosome occupancy of each transcript by ribosome profiling (Fig. 4d, and Supplementary Fig. 9b–d). In line with the absence of global effect of the lack of tRNA t$^6$A modification on protein synthesis (Fig. 4c, and Supplementary Fig. 9a), we did not observe global changes in ribosome occupancy of ANN codons, at the A-site (Fig. 4e, and Supplementary Fig. 9e).

Depletion of OSGEP led to marked protein aggregation in melanoma B16F10 cells, as a result of mistranslation (Fig. 1c, d). To evaluate the specific consequence of OSGEP loss on protein synthesis, we

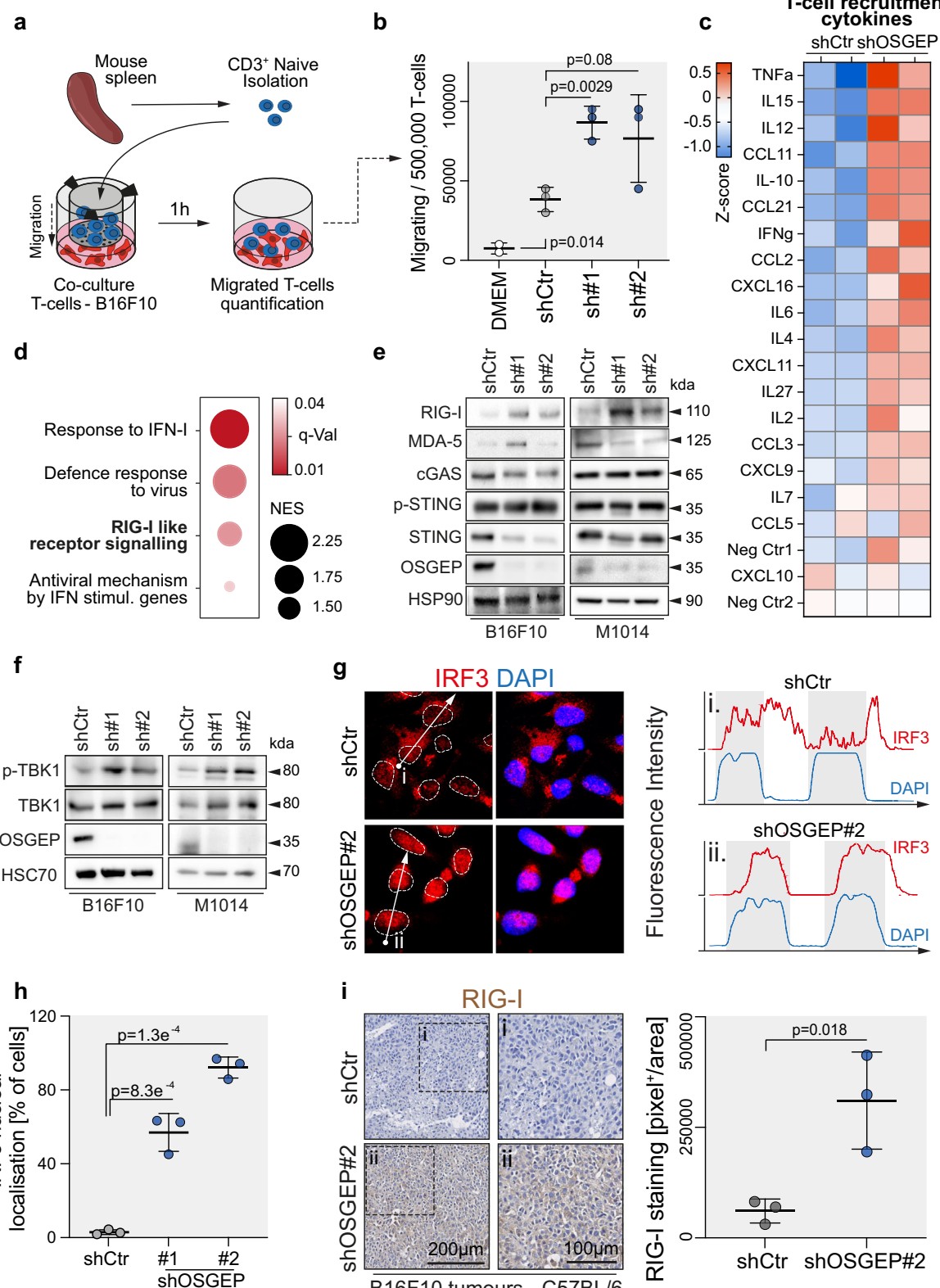

analysed the protein composition of purified aggregates by mass spectrometry (Fig. 4f, and Supplementary Dataset 4). Proteins detected in aggregates upon OSGEP depletion were involved in structural constituents of the ribosome, glutathione binding or unfolded protein binding, confirming that protein homeostasis was affected by the lack of t⁶A modification (Fig. 4g). Importantly, we calculated the average ribosome stalling (RPF count/gene) for each mRNA encoding proteins

found in aggregates in control and OSGEP knockdown cells. This analysis revealed a significant increase in ribosome stalling values for OSGEP-depleted aggregate mRNAs compared to controls (Fig. 4h). Moreover, the mRNAs corresponding to proteins uniquely found in aggregates of OSGEP-deficient cells exhibited increased ribosomal occupancy on ANN codons, particularly at the A-site, when compared to the ones uniquely found in shCtr cells or a random list of mRNAs

**Fig. 3 | OSGEP loss leads to activation of the RIG-I-dependent pathway.**
**a** Scheme of experimental procedure of naïve T cells isolated from mouse spleen to perform a migration co-culture assay in Transwell® with B16F10 cells.
**b** Quantification of the number of T cells migrating through Transwell® containing DMEM medium or shCtr or shOSGEP B16F10 cells ($n = 3$ Transwell® per condition).
**c** Cytokinome profiling of DMEM medium collected after 48 h from B16F10 cells infected with shCtr or shOSGEP ($n = 2$ technical replicates per condition). **d** Gene set enrichment analysis (GSEA) of the shCtr versus shOSGEP B16F10 cells focusing on inflammatory pathways (NES: normalized enrichment score). **e** Western blot for RIG-I, MDA-5, cGAS, STING, p-STING and OSGEP in B16F10 cells infected with control (Ctr) and OSGEP shRNAs (#1, #2). HSP90: loading control. **f** Western blot for TBK1, p-TBK1 and OSGEP in B16F10 cells infected with control (Ctr) and OSGEP shRNAs (#1, #2). HSC70: loading control. **g** Immunostaining (left) and fluorescence intensity (right) of IRF3 and DAPI in the outlined area (dotted square) in B16F10 cells control (shCtr) or depleted for OSGEP (shOSGEP#2). **h** Quantification of the percentage of cells with nuclear localization of IRF3 staining ($n = 3$ independent experiments). **i** Representative immunostaining and quantification of RIG-I staining in B16F10-derived tumours (day 14) infected with shRNAs Ctr or targeting OSGEP ($n = 3$ tumours per condition). Shown is mean +/- SD (**b**, **h**, **i**). Unpaired two-tailed $t$-test (**b**, **h**, **i**). Exact $p$-values are indicated. Source data are provided as a Source Data file.

(Fig. 4i–k, Supplementary Fig. 10a–c, and Supplementary Dataset 5 and 6). Similar to previous studies, knockdown of OSGEP affected differently ANN codons (Fig. 4l)[21,22,27]. Together, our data indicate that the lack of t⁶A affects the decoding of ANN codons at the A-site.

### OSGEP restricts activation of RIG-I by HRI-induced stress granules

When exposed to stress, such as a temperature change and infection, cells can activate the integrated stress response pathway (ISR) to minimise cellular damage. However, the ISR pathway can also be activated by the accumulation of misfolded proteins through Heme-Regulated Inhibitor (HRI), as a result of altered translation[52–55]. Since loss of t⁶A is responsible for an increase in protein aggregates, we tested whether it induced the ISR pathway activation. Depletion of OSGEP in B16F10 or A375 cells was responsible for a strong induction of HRI protein expression (Fig. 5a, and Supplementary Fig. 11a). In line with this, we also observed an increase in the phosphorylation of the main target of HRI, alpha subunit of eukaryotic translation initiation factor 2 in both cell lines tested, despite the absence of measurable global translation repression, suggesting that the ISR induction observed is modest and selective (eIF2α; Fig. 5b, and Supplementary Fig. 11b). Similarly, depletion of LAGE3 in B16F10 cells lead to the induction of both HRI and pEIF2α (Supplementary Fig. 11c, d). Of note, HRI was not induced by proteasome inhibitors (i.e. MG132, Bortezomib) or untranslated protein response (UPR) activator (i.e. Tapsigargin) in B16F10 cells, indicating that KEOPS deficiency activates HRI independently of these pathways (Supplementary Fig. 11e, f).

The ISR pathway activation further leads to the assembly of cytoplasmic stress granule (SGs), which serve as a protective mechanism of the mRNA translation process until resolution of the stress[53,54]. These granules house mRNAs, small RNAs, RNA-binding proteins and other components of the translation machinery[56]. First, using Ras-GAP SH3-binding protein 1 (G3BP1) as a marker of SGs assembly[57], we found that depletion of OSGEP in B16F10 cells triggered the formation of SGs (Fig. 5c, d, f, g). Second, in line with previous studies, we also found RIG-I colocalized in stress granules of cells lacking OSGEP[58–60] (Fig. 5d, e). To test whether SGs play a central role in the activation of RIG-I pathway upon depletion of OSGEP, we generated B16F10 cells carrying the depletion of both OSGEP and HRI. Strikingly, the depletion of HRI abolished the formation of stress granules observed in OSGEP-deficient cells (Fig. 5f, g) and prevented RIG-I upregulation in OSGEP-deficient cells (Fig. 5h). As a confirmation, we also treated B16F10 cells with an inhibitor of EIF2α phosphorylation (ISRIB)[61], which led to a decrease in stress granule formation in OSGEP-depleted cells (Supplementary Fig. 12a, b). Of note, staining of stress granules and protein aggregates in OSGEP-depleted B16F10 cells showed no colocalization (Supplementary Fig. 12c). Also, ISRIB treatment that blocks ISR activation and subsequent assembly of SGs failed to reduce protein aggregate formation, indicating that protein aggregates formed upon OSGEP loss precede the assembly of SGs (Supplementary Fig. 12d). Together, this reveals that loss of OSGEP drives stress granule formation and RIG-I activation via the activation of HRI pathway.

### t⁶A modification averts tRNA recognition by RIG-I

The major function of RIG-I in eukaryotic cells is to act as a sensor of viral infection and to activate the immune response. However, upon various cellular stresses, RIG-I can also recognise RNA from the host[62–64]. To assess which RNA ligand is responsible for RIG-I pathway activation following loss of OSGEP, we performed an RNA-immunoprecipitation (RIP) assay using a FLAG-tagged RIG-I construct in B16F10 cells depleted for OSGEP (Supplementary Fig. 13a). First, we isolated RNA bound to RIG-I in control or OSGEP-depleted B16F10 melanoma cells to transfect wild-type (WT) cells. RIG-I and TBK1 were activated in WT cells only upon transfection with RNA bound to RIG-I from OSGEP-depleted but not control cells, confirming the immunogenic potential of endogenous RNA resulting from OSGEP depletion (Fig. 6a). We subjected the immunoprecipitated RNA-RIG-I complex from shCtr or shOSGEP B16F10 melanoma cells to total RNA sequencing. In line with previous studies, RIG-I was observed to bind various types of endogenous RNA[65,66]. Strikingly, tRNAs were the most differentially enriched RNA species bound to RIG-I in response to OSGEP loss (Fig. 6b). Interestingly, this enrichment included tRNAs from terminal base anticodon groups (NNU, NNA, NNC, NNG), which were uniformly upregulated, regardless of their status as targets of t⁶A modification (Fig. 6c, d, and Supplementary Fig. 13b–d). Together, this suggests that the loss of OSGEP broadly impacts the recognition of tRNAs by the RIG-I sensor.

To confirm the pivotal role of RIG-I in the OSGEP-induced immune response, we generated B16F10 cells carrying depletion for both OSGEP and RIG-I, effectively disabling the downstream pathway of the latter (Fig. 6e). We then transplanted the double knockdown cells into immunocompetent mice. RIG-I loss, alone, was sufficient to completely restore the growth deficit of OSGEP-depleted tumours (Fig. 6f). Moreover, the infiltration and activation of CD4⁺ and CD8⁺ T-cells were also eradicated in double knock down tumours (Fig. 6g, h, and Supplementary Fig. 14a–f). These data show that RIG-I activation is essential to promote the T cell-dependent anti-tumoral effect seen upon OSGEP depletion.

In summary, our data reveal that OSGEP, and by extension t⁶A tRNA modification, prevents T cell-mediated immune response in melanoma by maintaining ANN codon decoding translation, which avoids activation of RIG-I in stress granules by tRNAs, thereby allowing tumour growth (Fig. 6i).

## Discussion

Here, we show that OSGEP, responsible for the t⁶A modifications on cytoplasmic tRNAs, directly contributes to tumour malignancy by ensuring tumoral T cell exclusion. While mRNA translation is known to be rapidly and dynamically regulated by tumour cells to accommodate their gene expression needs, the consequences of inefficient translation on the gene expression program of tumour cells remain largely unexplored. For instance, eIF4F and eIF5B translation machinery factors are required for the expression of PD-L1 in tumour cells[8,9]. The mRNA m⁶A modification is required to reduce immune cell infiltration in colon cancer by directly affecting the transcript stability and expression of STAT1 and IRF1[67]. However, it

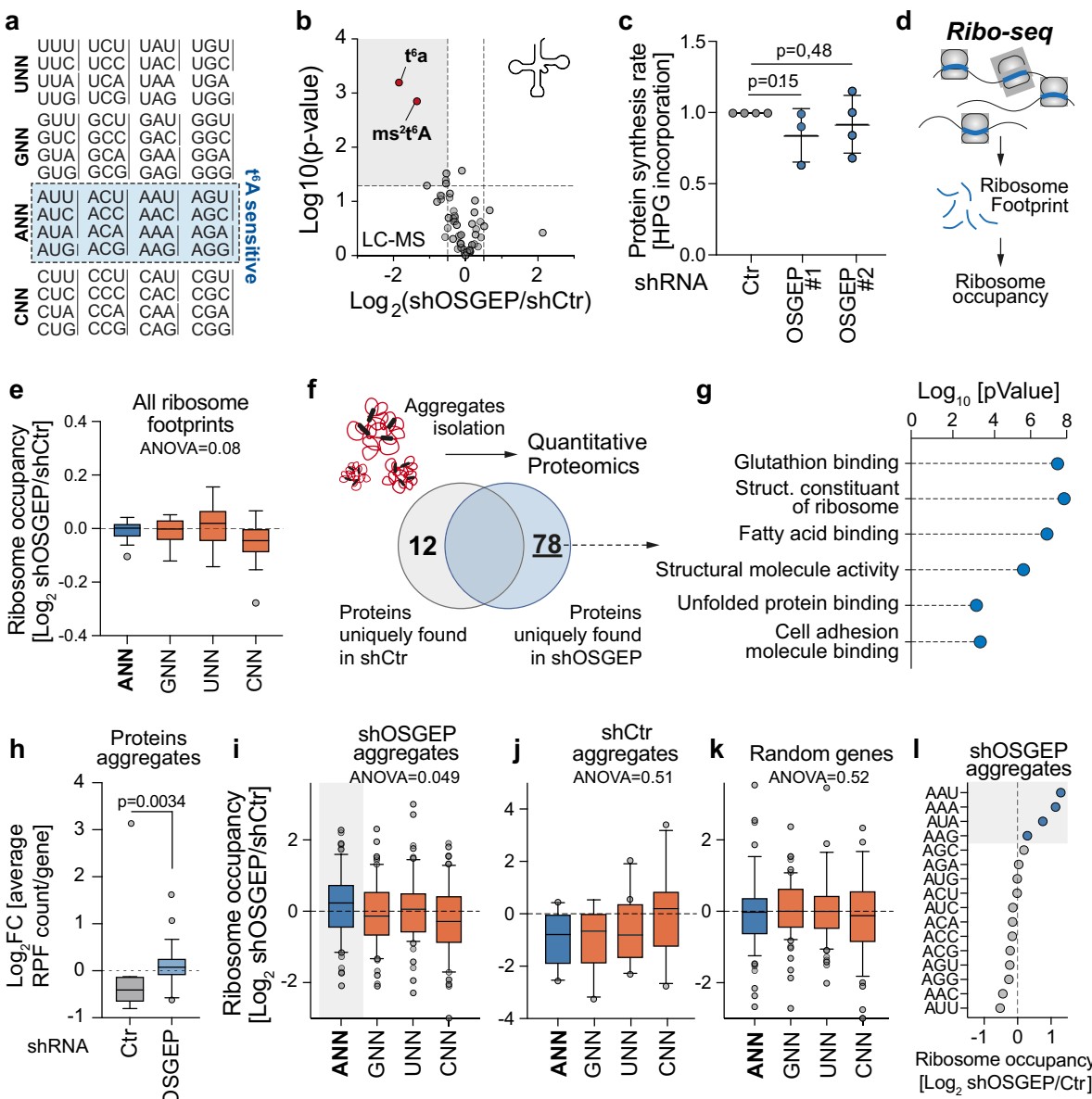

**Fig. 4 | tRNA threonylation is required for efficient cytoplasmic translation.**
**a** Codon chart representation with identification of the codons sensitive to t⁶A modification (ANN). **b** Volcano plot of the 52 most abundant modifications from extracted tRNAs of B16F10 cells infected with shRNA control (shCtr) or targeting OSGEP (shOSGEP) by LC-MS. **c** Quantification of protein synthesis in cells infected with control (Ctr) or OSGEP shRNAs (#1, #2) in B16F10 cells. shCtrl $n = 4$, shOS-GEP#2 $n = 4$, shOSGEP#1 $n = 3$ biological replicates from independent experiments. **d** Scheme of experimental procedure for ribosome profiling to determine ribosome occupancy on each transcript. **e** Quantification of fold change in A-site ribosome occupancy on ANN, GNN, UNN and CNN codons in B16F10 cells infected with shCTR or shOSGEP on 25547 genes detected. **f** Overlap and uniquely found proteins observed in aggregates of B16F10 cells infected with shRNA control (shCtr) or targeting OSGEP (shOSGEP). **g** Gene ontology analysis (ToppGene) of proteins uniquely found in aggregates of B16F10 cells infected with shRNA targeting OSGEP. **h** Ribosome stalling analysis performed with RiboDipa. Ribosome stalling (RPF count per gene) for each mRNA encoding proteins found in aggregates in shCtr ($n = 12$) and shOSGEP ($n = 78$) cells, expressed as log2FC (shOSGEP/ shCtrl) was calculated ($p$-value with Mann–Whitney test). Ribosomal occupancy on ANN, GNN, CNN, UNN codons of transcripts of corresponding proteins uniquely found in shOSGEP aggregates (**i**; $n = 78$), in shCtr (**j**; $n = 12$) and randomly selected transcripts (**k**; $n = 78$). **l** Ribosomal occupancy on the 16 ANN codons of transcripts of corresponding proteins uniquely found in shOSGEP aggregates. Box plots show 10th percentile, first quartile, median, third quartile, and 90th percentile (**e, i–k**). Shown is mean +/- SD (**c**). Unpaired two-tailed $t$-test (**b, c, g**), ANOVA Kruskal–Wallis test (**e, i–k**). Source data are provided as a Source Data file.

has become increasingly clear that successful immune evasion requires efficient mRNA translation not only to produce pro-immunogenic effectors, but also to prevent their activation[9,14]. Depletion of m⁶A enzyme, METTL3, induces the formation of non-standard dsRNA and deleterious immune response during hemato-poietic development[68]. Our work indicates that the lack of tRNA t⁶A modification is responsible for the accumulation of stress granules, which in turns activate RIG-I and initiate an anti-tumoral immune response in melanoma tumours.

A direct regulatory role for t⁶A tRNA modification in deter-mining RIG-I activation was unexpected. RIG-I was originally identi-fied as a viral short RNA sensor, which, upon activation, promotes the production of secreted type 1 interferon pro-inflammatory cytokines[28]. However, RIG-I was recently shown to also bind endo-genous RNA upon cellular stress, such as ROS production, hypoxia, or heat shock[62–64]. Our data reveal that melanoma cells lacking t⁶A modification on tRNAs induce ribosomal pausing on ANN codons of specific transcripts. The altered proteostasis triggers the formation

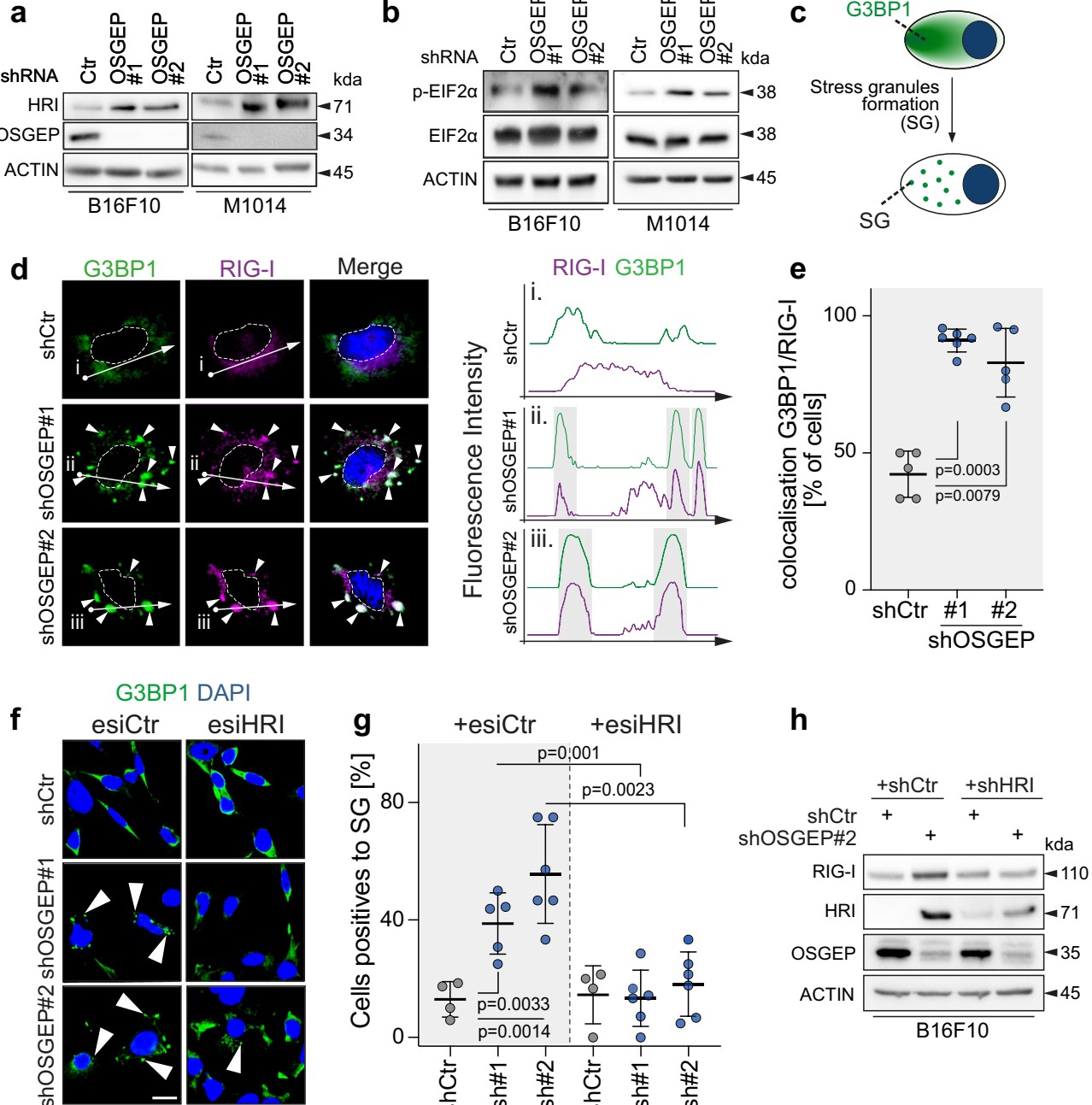

**Fig. 5 | OSGEP depletion activates RIG-I pathway via HRI-induced stress granules. a** Western blot for HRI in B16F10 cells infected with control (Ctr) and OSGEP shRNAs (#1, #2). ACTIN: loading control. **b** Western blot for p-EIF2α, EIF2α, in B16F10 and M1014 cells infected with control (Ctr) and OSGEP shRNAs (#1, #2). ACTIN: loading control **c** Scheme of cellular relocation of stress granule marker G3BP1 upon formation of stress granules. **d** Immunostaining (left) and fluorescence intensity (right) of G3BP1 and RIG-I in the outlined area (dotted arrow) in B16F10 cells control (shCtr) or depleted for OSGEP (shOSGEP#1, #2; DAPI is blue). **e** Quantification of the percentage of cells with colocalization of G3BP1 and RIG-I

staining ($n = 5$ different experiments). Fluorescence (**f**) and quantification (**g**) of G3BP1 staining in B16F10 cells infected with shCtr or shOSGEP, and treated with esiCtr or esiRNA targeting HRI. $n = 4$ independent pictures were quantified for shCtr, $n = 5$ or $n = 6$ for shOSGEP. **h** Western blot for OSGEP, RIG-I and HRI in B16F10 cells infected with control (shCtr) and OSGEP shRNAs (#1, #2), and double-depleted for HRI with shHRI. ACTIN: loading control. Shown is mean +/− SD (**e, g**). Unpaired two-tailed $t$-test (**e, g**). Exact $p$-values are indicated. Source data are provided as a Source Data file.

of stress granules, which in turn facilitates the activation of RIG-I by endogenous tRNA.

Disrupting codon-specific mRNA translation by depleting OSGEP leads to the generation of specific protein aggregates. HRI, one of the kinases of the ISR, is known to be activated upon proteotoxic stress[55]. Therefore, it is not surprising that HRI is activated and engages ISR activation and SG formation in cells lacking OSGEP. We show here that HRI is essential for RIG-I activation, through co-

localization with stress granules. RIG-I exhibits a marked increase in its binding to tRNAs following OSGEP depletion. Of note, we did not find binding specificity towards tRNA targets of t⁶A modification. This indicates that RIG-I is likely binding tRNAs as result of mis-translation and accumulation of translation machinery in stress granules. Similar to our findings in melanoma, several studies have suggested stress granules as a described concentrator of RNAs, examined by RIG-I RNA sensor[59,60,69]. How and under which

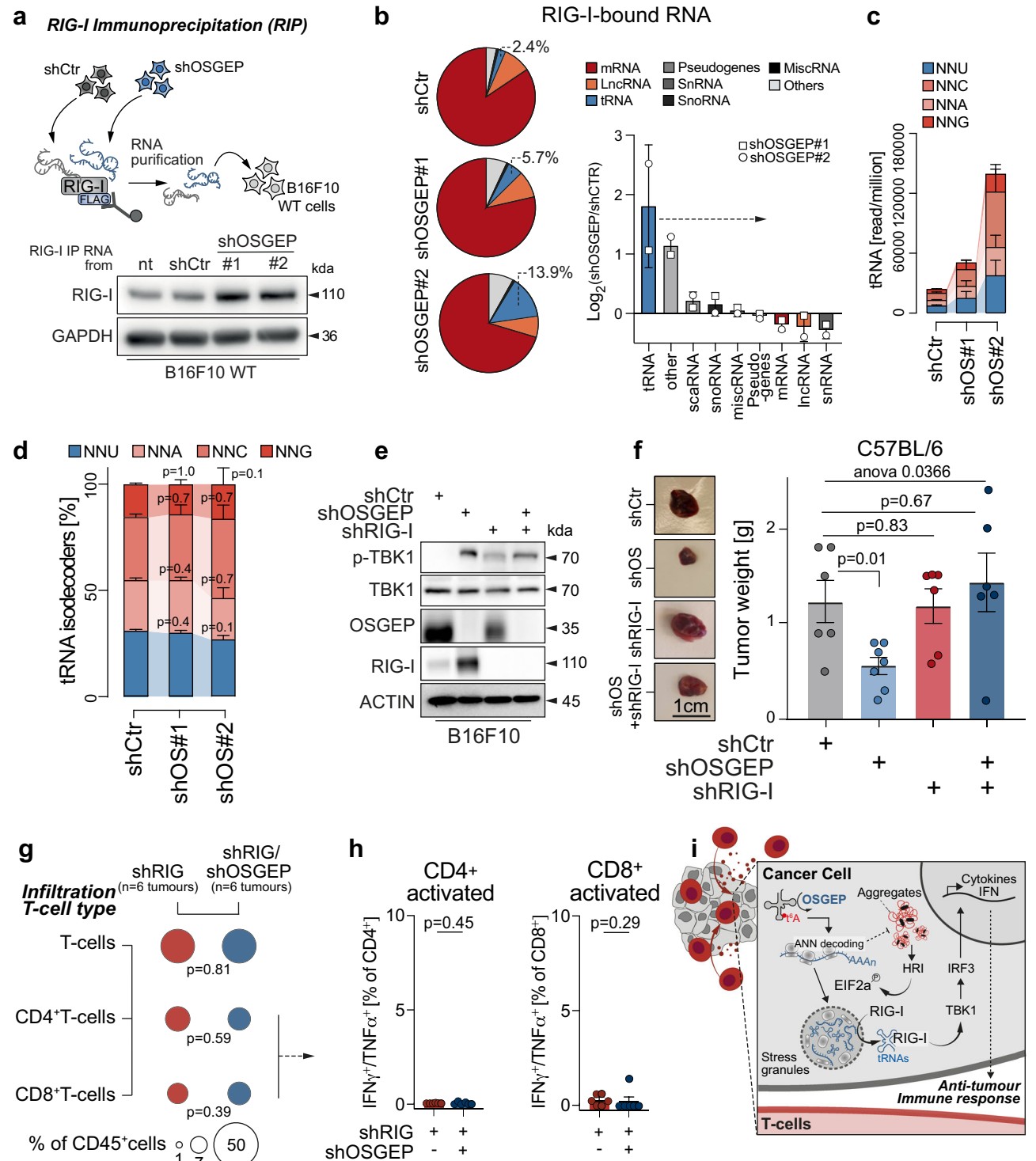

conditions does RIG-I bind tRNA molecules is an intriguing question that deserves further investigation.

Finally, we have uncovered the essential role of HRI in driving RIG-I activation. This highlights the potential of HRI modulators - originally associated with heme deficiency in erythroid cells - as promising strategies to enhance T cell recruitment and activity in cancer. By regulating RIG-I activity, these modulators may unlock unique clinical opportunities for cancer treatment.

We demonstrate that an OSGEP-driven transcriptome signature has predictive value in melanoma patients. Genes upregulated by OSGEP depletion, resulting in reduced t6A levels, correlate with T cell infiltration and poor overall survival. These findings suggest that t6A RNA modification regulators should be considered in the growing list of RNA-modifying anticancer drug targets.

However, the therapeutic strategy of inhibiting tRNA threonylation by targeting OSGEP may not be cancer-specific. Inhibiting the catalytic subunit of the KEOPS complex could also impact its other subunits, potentially causing broader effects. Long-term systemic inhibition may lead to side effects, as patients with loss-of-function mutations in KEOPS complex genes survive but exhibit combined neuronal and renal deficiencies[25,26].

**Fig. 6 | OSGEP loss prompts tRNA recognition by RIG-I. a** FLAG-RIG-I was immunoprecipitated from control (shCtr) or OSGEP-depleted (shOSGEP) B16F10 cells and Flag-RIG-I bound RNAs were purified and transfected or not (nt) into wild-type B16F10 cells. RIG-I, TBK1 and p-TBK1 expression is detected by western blot and GAPDH used as loading control. Quantification of the percentage bound (**b**) and differential binding (**c**) of RNA species to RIG-I resulting from FLAG-RIG-I RIP-seq in B16F10 cells infected with control (Ctr) or OSGEP shRNAs (#1, #2) (*n* = 2 independent shRNAs from 3 biological replicates from independent experiments). **d** Quantification of the differential binding of tRNA anticodon groups (NNU, NNA, NNG, NNC) to RIG-I by RIP-seq (*n* = 3 biological replicates from independent experiments). **e** Western blot for OSGEP, RIG-I, p-TBK1 and TBK1 in B16F10 cells infected with control (shCtr), RIG-I (shRIG) and/or OSGEP shRNAs (#1, #2). ACTIN: loading control. **f** Representative B16F10 tumours and quantification of tumour weight, 14 days after subcutaneous transplantation (*n* = 6 C57BL/6 mice per condition). Tumours were derived from B16F10 cells infected with control (shCtr), RIG-I (shRIG) or OSGEP (shOSGEP) shRNAs. **g** Bubble plot of the percentage of infiltrating T cell subpopulations normalised to CD45$^+$ cells (average of *n* = 6 shRIG and *n* = 6 shRIG/shOSGEP tumours). **h** Quantification of the percentage of CD4$^+$ (left) or CD8$^+$ (right) T cells IFNγ$^+$/TNFα$^+$ in B16F10 tumours, 14 days after transplantation in immunocompetent mice (shRIG *n* = 6; shRIG/shOSGEP *n* = 6 mice). **i** Proposed model. OSGEP facilitates the deposition of t6A on specific tRNAs and ensures accurate decoding of ANN codons. OSGEP loss triggers accumulation of protein aggregates and the activation of the HRI and EIF2α pathways, which drive SG assembly, RIG-I activation and an anti-tumoral T cell response in melanoma. Shown is mean +/− SD (**b**−**d**, **f**, **h**). Two-way ANOVA (interaction factor variation (**f**); Mann–Whitney test two-tailed (**d**, **g**, **h**); ANOVA Kruskal–Wallis two tailed test (**f**). Exact *p*-values are indicated. Source data are provided as a Source Data file.

Targeting the immune system has emerged as a standard cancer therapy. Inducing primary tumour cells to recruit and activate immune cells offers a promising approach to halt tumour growth and prevent relapse. We propose that inhibiting t$^6$A tRNA modification represents a therapeutic strategy to activate immune responses in 'cold tumours' and prevent cancer progression.

# Methods

## Ethics statement
Our research complies with all relevant ethical regulations. Our in vivo experiments are approved by the Ethical committee of the University of Liege (#2268). The study design followed the principles of the 3Rs (Replacement, Reduction, and Refinement) to ensure responsible animal use. The human study sample project was reviewed and approved by Research Ethics Committee and NHS Health Research Authority (IRAS project ID: 216310 REC reference: 16/LO/2098), sponsored by the University of Manchester

## Cell culture and reagents
All cells were cultured in a humidified incubator at 37 °C with 5% CO$_2$. Melanoma cell line B16F10 (ATCC CRL-6475), A375 (ATCC CRL-1619) and Lenti-X293 T (Sigma-Aldrich) were grown in Dulbecco's modified Eagle's medium (DMEM, LONZA) supplemented with 1% glutamine and 1% penicillin-streptomycin and 10% FBS (GIBCO). M1014 were obtained from Lionel Larue (Curie Institute), MM117 and MM011 were provided by Pr. G. Ghanem (Institut J. Bordet, Université Libre de Bruxelles)[45] and were grown in HAM's medium supplemented with 10% FBS and 1% penicillin-streptomycin. All cells were routinely tested negative for mycoplasma contamination.

Cells were treated with 5 μM of MG132 (Enzo, #MG132-B12N), 15 μM of ISRIB compound (Sigma-Aldrich #SML843-5MG) for 24 h. Cells were treated with thapisgagin (Sigma-Aldrich #T9033) and bortezomib (medchemexpress#HY-10227) for 18 h. shRNA sequences, primers sequences and references for antibodies are provided in Supplementary Dataset 7. All shRNA plasmids were obtained from Vectorbuilder. For double knockdown, B16F10 cells were first transduced with lentiviral shRNA CTR and shRNA RIG-I according to manufacturer's instructions. These cells were transduced again 3 days later with lentiviral shRNA CTR or OSGEP to achieve double knockdown. Knockdown efficiency was evaluated by RT-qPCR and/or by western blot analysis. For overexpression of RIG-I, lentivirus construct containing FLAG-tagged RIG-I was generated by VectorBuilder. All esiRNA sequences were purchased from Sigma-Aldrich/Merck. For transient transfections, a mix of 30 nM of esiRNA and DharmaFECT 2 transfection reagent (Horizon) were added to the medium for 6 h. Media were changed, and depletion was evaluated after 48 h by RT-qPCR. esiHRI transient transfection with pre-designed silencer (Invivogen #AM16708) was performed using the Lipofectamine®2000 reagent protocol.

## Lentiviral infections
Lenti-X 293T cells were transfected with 12 μg of the lentiviral construct of interest (shRNA or overexpressing plasmids), 5 μg of VSVG and 12 μg of pPsx2 using Mirus transfection reagent (Mirus ™ TransIT™-LT1 Transfection Reagent, MIR 2360, Fisher Scientific). All constructs are listed in Supplementary Dataset 7. After 48 h, the supernatant was collected, centrifuged to remove cell debris, filtered through a 0.45 μm filter and polybrene was added to the medium before transduction to target cells. The same procedure was repeated on the second day. Cells were then selected with puromycin (1 μg/ml) for 48 h. Knockdown or overexpression efficiency was confirmed by qPCR or western blot analysis. Three independent transductions were performed for all experiments.

## Mice and treatment
Female C57BL/6 and NOD-SCID N mice were obtained from Charles River. Female mice were 6–8 weeks old at the time of tumour inoculation due to ease of handling and housing logistics. No sex-stratified analyses were performed, as the primary focus of this study was the characterisation of tumour growth dynamics under the knockdown of our protein of interest. Future studies may explore potential sex-based differences in immune response to the lack of t$^6$A. All mouse husbandry and experiments were carried out according to the local ethics committee ULiege under the terms and conditions of the animal licence #2268. Mice were maintained under controlled environmental conditions with a 12 h light/12 h dark cycle, temperature of 22 ± 2 °C and relative humidity of 40–60%, with free access to food and water.

1 × 10$^6$ B16F10, M1014 cells resuspended in PBS were subcutaneously injected into the right flank of animals. Tumour size was measured once palpable by assessing the longest dimension (length) and the longest perpendicular (width). Tumour volume was estimated with the formula ($L$×$W^2$)/2. The maximal tumour size permitted by the ethical committee (1 cm$^3$) was not exceeded. For antibody treatments, antibodies were purchased from BioXcell: α-IgG (cat#BE0089) α-CD4 (cat#BE0119), α-CD8 (cat#BE0117), α-NK1.1 (cat#BE0036). Mice were treated with 500 μg, 2 days prior tumour inoculation, followed by 250 μg at days 3, 8 and 13 post-tumour injection via intraperitoneal injection.

## Tumour preparation for flow cytometry
Tumours were extracted on day 14 post injection and dissociated in digestion media (collagenase, FBS, DNAse) with GentleMACS using digestion tumour program mTDK2, 37 °C for 45 min. Staining was performed according to protocols of transcription factor, intracellular and cell surface staining from Milteny's manufacturer. Briefly, tumours were filtered through 70 μM cell strainer and cells were counted. 1,000,000 tumour cells were stained with 7-AAD viability dye to exclude dead cells, washed and stained with cell surface antibodies. INFγ and TNFα staining was performed using Inside Stain kit (Miltenyi #130-090-47), according to manufacturer's instructions. Briefly, cells

were fixed and permeabilized with Fixation/Permeabilization solution during 30 min, and stained with the mix of antibodies for 30 min in the dark at 4 °C.

All cells were washed, resuspended in FACS staining buffer for analysis on BD FACSCanto™ II. Data were analysed using FlowJo Software (Version 10). Antibodies are listed in Supplementary Dataset 7.

## Migration assay

Spleen from 7 to 12-weeks old mice was taken and smashed manually, and T cells were isolated using Pan T Cell Isolation Kit II for mouse from Miltenyi. Briefly, splenocytes were incubated first with Biotin-Antibody cocktail for 5 min at 4 °C and then mixed with Anti-Biotin Microbeads for 10 min at 4 °C. Magnetic cell separation was performed, and negative selection containing T cells was collected. For the migration assay, $5 \times 10^5$ T-cells in 300 μL of PBS were placed in the upper chamber of Transwell® Permeable support, 6.5 mm insert (24-well plate). In the lower chamber, $6 \times 10^4$ B16F10 cells were seeded for 48 h in 500 μL of supplemented DMEM. After 1 h, Transwell® were carefully removed, and pictures of the lower part were taken using Leica DMIL microscope and LASV4.11 software. Counting was performed manually using Thoma cell counting chamber and final count was calculated using the formula: Cell count=Counted cells $\times 10^4$ cells/ml.

## Quantification of protein aggregates

Protein aggregation was monitored by FACS using the PROTEOSTAT Aggresome detection kit (ENZO ENZI-51035-K100), according to the manufacturer's instructions. Briefly, cells were seeded in 6-well plates; MG132 treatment was used as positive control (5 μM, 18 h). Cells were then collected, fixed in 4% paraformaldehyde solution for 30 min, and permeabilized with 0.4% Triton X-100 for 20 min. Aggregation dye was added in 1:5000 dilution in the 1x Assay buffer for 30 min. Fluorescence was measured by BD FACSCantoII using L1 laser and the mean fluorescence intensity (MFI) was obtained. APF (Aggregation Propensity Factor) was calculated according to the formula APF = 100x(MFIko-MFIcontrol)/MFIko.

## Protein extraction and immunoblot assay

Cells were washed two times with PBS to ensure complete removal of media. On ice, cells were lysed with RIPA lysis buffer (25 mM Tris-HCl pH8.0, 150 mM NaCl, 1% NP-40, 1% sodium deoxycholate, 0.1% SDS) supplemented with Complete™ proteinase inhibitor cocktail (ROCHE, cat#04693116001) and phosphatase inhibitor (Roche, cat#04906837001). Cell lysates were incubated on ice for 15 min before centrifuged at top-speed during 10 min at 4 °C. Protein determination was assessed using BCA (Bicinchoninic Acid) protein Assay Kit from ThermoFisher (cat#23227) and according to manufacturer's instructions. Volume and quantity of proteins were adjusted and Laemmli buffer x5 was added, following boiling at 95 °C for 5 min. Samples were loaded on SDS-PAGE gels, followed by transfer on PVDF membranes (Milipore, cat#IPVH85R). Membranes were blocked in 5% milk at room temperature for 1 h and incubated overnight at 4 °C with the appropriate antibody. Next day, membranes were washed three times with TBST and incubated with mouse or rabbit secondary HRP antibodies in 5% milk at room temperature for 1 h. Membranes were washed three times with TBST, and ECL was applied to reveal them using Amersham ImageQuant 800. Pixel density was calculated using ImageJ measurement tool.

## Immunostaining

For staining of B16F10 cells in vitro, cells were fixed on a coverslip in 4% paraformaldehyde and permeabilized in PBS supplemented with 0.3% of Triton X-100. Blocking was done with 5% FBS for 1 h. Primary antibody was kept overnight at 4 °C in PBS supplemented with 0.3% Triton X-100 and 2% of FBS. Appropriate secondary antibody (dilution 1/1000) was added for 1 h and DAPI (1/10,000) for 15 min at room temperature in PBS with 0.3% Triton X-100 and 2% FBS. Images were acquired by confocal microscopy using Nikon A1R.

Extracted B16F10 subcutaneous tumours were fixed overnight with 4% paraformaldehyde, transferred to 70% EtOH and embedded in paraffin. Sections were deparaffinised and subjected to heat-induced antigen retrieval in 10 mM sodium citrate buffer (pH6). Then, slides were permeabilised for 10 min with PBS containing 0.3% Triton X-100 at room temperature and washed three times for 5 min in PBS. To block non-specific antibody binding, sections were incubated with blocking buffer comprising 2.5% goat serum (ImmPRESSTM kit, Vector Laboratories) in PBS with 0.1% Tween-20 (PBST) for 30 min. Tissue slides were incubated with primary antibodies overnight at 4 °C. To detect specific proteins of interest, slides were then incubated with primary antibodies diluted in 1% FBS in PBST at 4 °C overnight. Slides were then washed three times in PBS for 5 min each. To label the detected proteins, slides were incubated with the Alexa Fluor 488 and Alexa Fluor 568-secondary antibodies diluted in 1% BSA in PBST for 1 h at room temperature, protected from light (1:1000; ThermoFisher). Sections were washed as before and their nuclei counterstained with DAPI (1:10,000 in PBS; Sigma-Aldrich) for 10 min. Finally, sections were rinsed with PBS, and the glass coverslips mounted using fluorescence mounting medium (S302380-2; Agilent). The primary antibodies used were CD3 (1:200, #ab11089, abcam), RIG-I (1:100, #700366, Invitrogen), and IRF3 (1:100, #SC-9082, Santa Cruz), G3BP1 (1:250, # 66486-1-Ig, Proteintech).

For histological analysis on murine tissues, tissue sections (4 μm thick) were deparaffinized and rehydrated. For immunostaining, tissue sections were deparaffinized, rehydrated, and subsequently subjected to heat-induced antigen retrieval depending on the antibody. Endogenous peroxidase activity was inactivated with 3% $H_2O_2$. After blocking in 2.5% goat serum or BSA 3%, 0.01% Triton-X100 for 30 min at room temperature, tissue slides were incubated with primary antibodies overnight at 4 °C. Then, corresponding secondary antibody reagents directed against mouse or rabbit were used for detection. DAB (3,3-Diaminobenzidine) or AEC substrate was used as chromogen.

Whole slides were digitally scanned using a NanoZoomer 2.0 HT (Hamamatsu).

## Spatial transcriptomics

Indexed sequencing libraries were prepared from H&E-stained Visium Spatial Gene Expression Slides using the Visium Spatial for FFPE Gene Expression Kit, Human Transcriptome (10× Genomics, 1000338) according to the manufacturer's protocol. Library quality was checked using the Fragment Analyser (Agilent). Libraries were quantified by qPCR using a KAPA Library Quantification Kit for Illumina (Roche, 07960336001). Paired-end sequencing with read lengths of 28 + 10 + 10 + 50 bp was performed on a NovaSeq 6000 sequencer (Illumina Inc.). Data was processed through the standard Space Ranger pipeline.

Deconvolution analysis for cell type signatures in tumours was performed with the STdeconvolve package (version 1.4.0) standard workflow in R (version 4.3.0). This package performs a reference-free cell type deconvolution analysis using latent Dirichlet allocation (LDA) modelling to determine the optimal number of cell signatures. For analysis, spots with fewer than 100 transcript counts were removed, and genes that are expressed in 100% of spots or present in fewer than 5% of spots were removed. To find the optimal number of cell type signatures (topics) for each sample, a range of LDA models were fitted with between 10 and 35 cell types. The optimal number of cell type signatures (k) were determined by the model with the lowest perplexity. Marker genes for each signature were determined as the top 20 genes with the highest log2-fold change compared to other signatures (Supplementary Fig. 6c, d).

The filtered Space Ranger output was further processed and analysed in the Seurat package (version 5.1) in R. All samples were

analysed with the standard Seurat spatial transcriptomics workflow individually. Firstly, spots with fewer than 100 transcripts detected were removed from samples and cell type signature (topic) percentages from STdeconvolve analysis were added as to object metadata. Cell signatures of similar cell types were added together to form main tissue signatures (melanoma, immune, endothelial, stromal, hair follicle). Each sample was normalised with the SCTransform function. T-cell score and OSGEP gene signatures were calculated using the AddModuleScore function. To calculate the distance of each spot to T cell-enriched spots, the $x$ and $y$ co-ordinates were extracted with the GetTissueCoordinates function. For each sample, the T-cell score was grouped into quartiles, and the spots in the highest quartile were considered T-cell-enriched. For all spots, the minimum distance to any spot enriched for T-cells was calculated by computing the Euclidean distance from each spot to each T-cell spot and recording the smallest distance.

To directly compare T-cell and OSGEP signature expression between samples, the two samples were first normalised with SCTransform before merging together. PCA was performed on the merged object, and batch effects between samples were corrected using RunHarmony function in the harmony package (version 1.2.0). Dimensionality reduction and clustering were performed on the harmony reduction.

### Cytokinome analysis

Cytokine array was performed using Proteome Profiler Mouse XL (ARY028 R&D systems) and performed according to manufacturer's instructions. In brief, cytokine membranes were activated using Array buffer. Cell supernatants were collected and incubated with cytokine membranes overnight at 4 °C on rocking platform. Membranes were washed 3 times, and Detection Antibody Cocktail was added for 1 h at room temperature. Membranes were washed 3 times and Streptavidin-HRP was added for 30 min at room temperature. Membranes were washed 3 times, Chemi Reagent Mix was added and followed by revelation using ImageQuant LAS4000mini. Dot intensities were analysed using the Protein Array Analyser plug-in for ImageJ and following manufacturer's instructions.

### Protein synthesis rate quantification

Cells were plated into 6-well plates in complete DMEM media for 48 h. To assess translation activity, Click-IT Plus HPG Alexa Fluor 488 assay kit was used (#C10456, ThermoFisher). Briefly, L-homopropargylglycine was added to fresh media at 10 µM concentration for 30 min. Cells were trypsinised, washed with PBS and fixed with the Click-IT fixative. Pellets were resuspended with a mix of saponin-based permeabilization buffer and the Click-IT Plus reaction cocktail containing the fluorescent picolyl-azide dye, washed with the washing reagent and resuspended in the saponin-based permeabilization buffer. The analysis and quantification of the AlexaFluor 488 fluorescence intensity was performed on BD FACSCantoII.

### RNA sequencing

B16F10 cells were depleted for OSGEP using shRNA, with scramble shRNA serving as a control. Total RNA from three independent infections was extracted using TRIzol (Life Technologies) following the manufacturer's protocol. RNA integrity was assessed on the Bioanalyzer 2100 with RNA 6000 Nano chips, and all samples had RIN scores greater than 9. Libraries were prepared from 500 ng of total RNA using the Illumina Truseq RNA Sample Preparation kit V2. PolyA RNA was purified with polyT-coated magnetic beads, chemically fragmented, and used as templates for cDNA synthesis with random hexamers. The cDNA ends were then end-blunted, adenylated at the 3'OH ends, and ligated to indexed adaptors. The adaptor-ligated library fragments were enriched by PCR according to Illumina's protocol and purified. Libraries were validated on the Bioanalyzer DNA 1000 chip and

quantified by qPCR using the KAPA library quantification kit. Sequencing was performed on a HiSeq2000 with a paired-end 2 ×100 base protocol. Sequencing data from total RNA sequencing were preprocessed and aligned using the nf-core/rnaseq workflow (v.3.0). Briefly, a quality trimming cutoff of 10 was applied and adapters were removed automatically using TrimGalore. Pre-processed reads were aligned to the *Mus musculus* (GRCm38) reference genome using STAR aligner (v.2.6.1 d) and quantification was performed by Salmon (v.1.4.0) [2] using the star_salmon option. Differential expression analysis was performed on the triplicates of two groups (shCTR and shOSGEP) using the packages edgeR (v.3.14) [3] and DESeq2 (v.1.30) [4] in the R environment. To identify the Differentially Expressed Genes (DEGs), the Log2 Fold-Change (log2FC) and $p$-values were calculated, shCTR was used as the reference. DEGs were considered up- or down-regulated according to the sign of the log2FC.

A Log2(foldchange) value superior to 0.5 and inferior to −0.5, and $p$-value < 0.05 were defined as thresholds to select differentially expressed genes for further gene ontology analysis, using Toppgene software.

### Protein aggregate proteomics

Cells were pelleted and washed two times with cold PBS. Pellets were lysed using cytoplasmic buffer solution (340 mM sucrose, 3 mM of CaCl2, 0.1 mM of EDTA, 2 mM MgCl$_2$, 1 mM DTT, 0.5% NP-40 and protease inhibitor in Tris-HCl pH8) and incubated 10 min on ice. Lysates were centrifuged at 4 °C, 15 min at 1300 × $g$ and the collected supernatants were again centrifuged at 4 °C, 10 min at 20,000 × $g$. Supernatant of this fraction contains the cytoplasmic fraction and is stored at −80 °C. Pellets were washed with wash buffer (Na-phosphate pH6.8, PMSF 1 mM, 2% of NP-40 and protease inhibitor in distilled water).

After aggregates isolation, pellets were resuspended in 50 mM Tris buffer containing 8 M urea, reduced, alkylated and digested using Lys-C/Trypsin mix (Promega) for 4 h at 37 °C, then the urea concentration was lowered to 1 M by dilution with 50 mM Tris buffer and digested overnight. The digestion was stopped by adding trifluoroacetic acid and half of the protein digest was purified using Pierce C18 Tip 10 µL bed (Thermo Scientific) and dried. Peptides were suspended in 0.1% TFA and spiked with a commercial mixture of protein digest standards originated from non-human biological material: the MassPREP™ Digestion Standards (Waters), at 50 fmol of ADH per injection. This commercial standard consists of two standard mixtures (MPDS Mix 1 and MPDS Mix 2) containing protein digests of Yeast Alcohol Dehydrogenase (ADH), Rabbit Glycogen Phosphorylase b, Bovine Serum Albumin, and Yeast Enolase, present at known protein ratios, allowing for the relative quantitation of the spiked samples. Peptides were independently analysed by LC-ESI-MS/MS on an Acquity M-Class UPLC (Waters), hyphenated to a Q Exactive Plus (Thermo Scientific), in nanoelectrospray positive ion mode. The trap and analytical columns were respectively Symmetry C18 5 µm (180 µm × 20 mm) and HSS T3 C18 1.8 µm (75 µm × 250 mm; both Waters). The samples were loaded at 20 µL/min on the trap column in 98/2 water 0.1% formic acid (A)/acetonitrile 0.1% formic acid (B) during 3 min and subsequently separated on the analytical column at a flow rate of 600 nL/min with a linear gradient of 150 min (0 min, 98/2 A/B v/v; 5 min, 93/7; 135 min, 70/30; 150 min, 60/40). The total run time was 180 min. TopN-MSMS method, where $N$ was set to 12 was employed with resolution of 70,000 in MS, AGC target of 1e6 or maximum injection time of 50 ms and a mass range from 400 to 1600 $m/z$. The parameters for MS2 spectrum acquisition were resolution of 17,500 with AGC target of 1e5 or Maximum injection time of 50 ms and isolation Window of 2.0 $m/z$ with collision energy (NCE) of 28.

MaxQuant v 1.6.17.0 was used to perform peptide and protein identification against the SwissProt database limited to Mus Musculus taxonomy (17038 sequences, June 2020) and relative quantification

using LFQ normalization [J. Cox, M.Y. Hein, C.A. Luber, I. Paron, N. Nagaraj, M. Mann, Mol. Cell. Proteomics: MCP, 13 (2014), pp. 2513–2526]. Oxidation of methionine and deamidation of glutamine and asparagine were set as variable modifications, carbamidomethylation of the cysteines as fixed modification, peptide spectrum match and protein false discovery rates (FDR) were both set at 0.01.

## Small RNA sequencing

For small RNA sequencing, cells were pelleted, and RNA was isolated using TRIzol Reagent according manufacturer's instruction. RNA was sent to ArrayStar for isolation, sequencing and analysis. Briefly, RNA samples were quality controlled for quantity by NanoDrop ND-1000 spectrophotometer and for RNA integrity by Bioanalyzer 2100 or gel electrophoresis. For each sample, 100 ng of total RNA was firstly dephosphorylated to form the 3-OH end. The 3-OH-ended RNA was then denatured by DMSO and enzymatically labelled with Cy3. The labelled RNA was hybridized onto Arraystar Mouse small RNA Microarray (8x15K, Arraystar) and the array was scanned by an Agilent Scanner G2505C. Agilent Feature Extraction software (version 11.0.1.1) was used to analyse acquired array images. Quantile normalization and subsequent data processing were performed using GeneSpring GX v12.1 software package (Agilent Technologies). After normalization, the probe signals having Present (P) or Marginal (M) QC flags in at least 1 out of 6 samples were retained. Multiple probes from the same small RNA (miRNA/tsRNA (tRF&tiRNA)/pre-miRNA/tRNA/snoRNA) were combined into one RNA level. Differentially expressed small RNAs between two comparison groups were identified by fold change (FC) and statistical significance (p-value) thresholds. Hierarchical Clustering heatmaps, scatter plots and volcano plots were plotted to display small RNA expression patterns among samples by R software.

## RNA modifications quantification (LC-MS)

For LC-MS-based tRNA modification analysis, cell pellets were sent to ArrayStar for tRNA isolation and sequencing. Briefly, tRNAs were isolated from total RNA checked for quality control by Urea-PAGE electrophoresis. 60–90 nt bands of tRNAs were excised and purified by ethanol precipitation. Purified tRNAs were quantified using Qubit RNA HS Assay kit (ThermoFisher, Q32855). tRNAs were hydrolysed to single nucleosides, and then nucleosides were dephosphorylated by enzyme mix. Pretreated nucleoside solution was deproteinized using Sartorius 10,000-Da MWCO spin filter. Analysis of nucleoside mixtures was performed on Agilent 6460 QQQ mass spectrometer with an Agilent 1260 HPLC system. Multi-reaction monitoring (MRM) mode was performed. LC-MS data was acquired using Agilent Qualitative Analysis software. MRM peaks for each modified nucleoside were extracted and normalized to the quantity of purified tRNAs.

## Ribosome profiling

Ribosome profiling was performed using Immagina Biotechnology's RiboLace kit for ribosome pulldown and LACEseq kit for library preparation. Briefly, B16F10 cells depleted or not for OSGEP were grown to confluence in a 10 cm dish and treated with 10 μg/mL cycloheximide (CHX) for 5 min at 37 °C to arrest translation, followed by cell lysis and RNA digestion. Active ribosomes were then pulled down using a puromycin analogue (RiboLace probe) and magnetic beads, with two pulldowns performed for each sample. RNA was subsequently extracted with the acid phenol-chloroform method, followed by size selection of 25–35 bp fragments in a 15% TBE-urea PAGE gel using Immagina's PAGExt gel extraction system. RPFs were quantified with the Qubit™ microRNA Assay Kit before proceeding with the library preparation.

RPF library preparation with LACEseq involved sequential 5′ phosphorylation, ligation, and circularization steps, with small RNA purification after each step using the RNA Clean & Concentrator™−5 kit from Zymo Research. Reverse transcription and an initial PCR

amplification were then performed, followed by cleanup of the PCR product using AMPure XP beads. A second PCR amplification was carried out to incorporate unique dual indexes using Immagina's iUDI Plate. Final cleanup of the indexed library was done using the NucleoSpin Gel and PCR Clean-Up kit followed by size selection in a 10% TBE PAGE gel with PAGExt gel extraction. Library quality and size were confirmed on a Bioanalyzer 2100, and sequencing was conducted on a NovaSeq 6000 (Illumina) at the NGS platform of the GIGA Institute, with 70 million reads per sample. Sequencing data from ribosome profiling were pre-processed by trimming the LACE-seq 3′-end linker, trailing 5′ (NNNNT) and 3′-end (NNNN) degenerated sequences using Cutadapt [1] (Trim Galore wrapper v.0.6.6). Reads within a 27–33 nucleotide range (estimated by Immagina) were selected using SeqKit tool (v.2.1.0) and aligned to the Mus musculus coding transcriptome (GRCm38 - v41) using STAR aligner (v.2.7.9a). Only reads with a perfect match were kept for further analysis. On average, 3.8 million reads per sample were mapped uniquely.

The optimal offset P site was determined from the 5′-ends of the selected reads and deduced from the location of the translation initiation site. The P site offset is defined specifically by looking at the global distribution of CDS start codon position (Methionine – AUG) and taking the most frequent position. We therefore selected reads with 0 and −1 frames of coding sequences and used a deduced offset P site of 15nt from the 5′-end of the reads (methodology from ref. 36).

After identifying the three ribosomal sites positions in the reads, the 5′-end has been trimmed to keep only the E, P, A sites (also known as Ribosome Protected Fragments) and 3′-end flanking regions of the reads. Flanking regions are the three non-tRNA binding codons downstream of the A site, used as non-translated control regions for the codon occupancy calculation.

To assess the ribosome occupancy, the frequency of each of the codons in the ribosomal sites and flanking regions was calculated. The ribosome occupancy metric is the result of the division of the frequency of the codons in the given ribosomal site by the average frequency in the flanking regions. The average ribosome occupancy by codon or of codons by transcripts subpools was extracted.

The ribosome stalling analysis has been performed using the R package RiboDiPA on the ribo-seq triplicates of the two conditions (shCtr and shOSGEP)[70]. Transcripts were divided into genomic locations bins. To obtain the genomic locations of the ribo-seq data, we re-mapped the RPFs data to the Mus musculus (GRCm38) reference genome using STAR aligner (v.2.6.1d) and used mapped reads BAM files for the analysis. In the context of this study, we assessed, for each gene of interest, the ribosome stalling (number of reads mapping) at each location and calculated an average gene ribosome stalling by pooling altogether these locations values. To identify a change in the ribosome stalling, the Log2 Fold-Change was calculated for each gene, shCtr being used as the reference.

All the steps described to perform the ribosome occupancy calculations were implemented in custom python (v.3.8.1) and bash scripts.

## Polysome profiling

B16F10 melanoma cells, depleted or not for OSGEP were gently scraped in phosphate-buffered saline containing 100 μg/ml cycloheximide on ice. After centrifugation at 200 × g for 5 min at 4 °C, the cells were lysed in hypotonic buffer containing 2.5 mM MgCl2, 5 mM Tris pH 7.5, 1.5 mM KCl and 1× protease inhibitors. Subsequently, CHX, dithiothreitol, RNAse inhibitor and Triton X-100 followed by sodium deoxycholate were sequentially added to solubilize cytosolic and endoplasmic reticulum-associated ribosomes. The lysates were then centrifuged at 16,000 × g for 7 min. The cytosolic lysates were loaded onto a non-linear sucrose gradient (from 5% to 34% to 55%) and ultracentrifuged at 220,000 × g for 2 h at 4 °C, using the SW60 rotor in an Optima XPN-80 ultracentrifuge. RNA abundance was measured

throughout the polysome gradient using a UV detector (Teledyne Isco), with detection at 254 nm.

## RIG-I/RNA complex immunoprecipitation

B16F10 cells with stable expression of FLAG-tagged RIG-I and of shRNA Ctr, OSGEP#1 or OSGEP#2 were grown on 100 mm culture dishes at 80% confluency. Cells were washed three times with cold PBS (Phosphate-Buffered Saline) and lysed in RIP buffer (150 mM KCl, 25 mM Tris pH 7.4, 5 mM EDTA, 0.5 mM DTT, 0.5% NP40) supplemented with protease inhibitors for 30 min at 4 °C. Lysates were cleared by centrifugation at $10,000 \times g$ for 15 min at 4 °C. 1 mg of lysates was mixed with anti-FLAG, anti-A/G protein-conjugated magnetic beads (MedeChemExpress, #HY-K0207, #HY-K0202 respectively) according to manufacturer's instruction. The binding reaction was incubated on rotation overnight at 4 °C. Beads were washed 4 times with RIP buffer and elution was completed with buffer elution A (0.15 M glycine, pH3). After 10 min, neutralization buffer (1 M Tris-HCl pH 8.0) was added, followed by incubation with 10 mg/ml of proteinase K (FischerScientific, #10103533) at 55 °C for 1 h. The RNA bound to FLAG-RIG-I was extracted using TRIzol protocol (Invitrogen). 1 µg/mL of RNA extracted from shCtr or shOSGEP depleted cells was transfected back on wild-type B16F10 using the lipofectamine 2000, according to manufacturer's protocol.

Library preparation was performed using 10 ng of total RNA following the Illumina Stranded Total RNA Ligation with Ribo-Zero Plus protocol (Illumina). The final library profiles were assessed using the QIAxcel Advanced System (Qiagen), and concentrations were measured by qPCR using the SYBR Fast KAPA Kit. Indexed Libraries were pooled in equimolar ratios, and the pooled library was sequenced on the NovaSeq 6000 platform using an S2 flow cell with paired-end 150 bp reads, with 50 million fragments sequenced per sample.

Raw reads were demultiplexed and adapter-trimmed using Illumina bcl2fastq conversion software v2.20. They were processed within our custom NextFlow RNAseq pipeline inspired from nf-core/rnaseq. We used TrimGalore (v0.6.7) to trim a single base at the 5' and 3' ends of each read and to perform a quality-based trimming after removing poly-G tails. Then, we did the STAR (v2.6.1d) alignment and Salmon quantification with the Mouse GRCm39 reference genome and the annotations from Ensembl release 112. Quality was assessed by MultiQC (v1.9) report. Raw counts were analysed using the DESeq2 package (v1.44.0) in R environment (v4.4.2), with the default methodology. Differential expression analysis was conducted, and pairwise comparisons were made between experimental conditions (e.g. shCtr vs. shOSGEP1). Results were filtered for adjusted $p$-values < 0.01. Gene biotypes were annotated using Ensembl's BioMart database. Statistical tests and results were conducted using the build-in DESeq2 functions.

Unmapped sequences from RNA sequencing were mapped to the *Mus musculus* mature tRNA reference genome (GRCm39) obtained from the Genomic tRNA database (GtRNAdb 2.0). Due to computing limitations, each sample was randomly subsampled to one million sequences using the sample function of the Seqtk tool (v.1.2). The mapping on the tRNAs has been performed using the STAR tool (v.2.7.10a), following Alex Dobin's recommendations regarding tRNA alignment. Identification of the mapped tRNA isodecoders mapped was done with a custom Python script (v.3.10.13) using the Pysam (v.0.22.0) library. Briefly, after retrieving the *Mus musculus* tRNA isodecoder set from GtRNAdb, every mapped was recognized and counted. To homogenize the results, the percentage of tRNA isodecoders per sample was calculated from the counts.

## In silico analyses using TCGA-HNSC datasets

RNA expression levels, clinical as well as follow-up data were downloaded from the TCGA-SKCM ($n = 473$) cohort (https://portal.gdc. cancer.gov/) in November 2019. For clustering analysis, expression values of the top 100 differentially expressed genes from RNA-seq of shCtr and shOSGEP B16F10 cells were ln(x+1) transformed and clustering of TCGA patients was performed using Euclidean distance and Ward (unsquared distances) linkage using Clustvis 2.0. Heatmaps with hierarchical trees were generated by the web tool ClustVis[58]. The infiltration score of CD4+ and CD8+ cells in SKCM-TCGA tumours was obtained from the TIMER score database (http://timer.cistrome.org).

## Sample sizing and collection

Graphs were generated using Prism 10 (GraphPad) or R studio (2023.09.1 + 494). No statistical methods were used to predetermine sample size, but at least three samples were used per experimental group and condition. Data distribution was tested for normality before using parametric tests. The number of samples is represented in the graphs as one dot per sample. If no individual samples are shown more than 5 samples per condition were used. Samples and experimental animals were randomly assigned to experimental groups. Sample collection was also assigned randomly. Sample collection and data analysis were performed blindly whenever possible. Whenever possible, automated quantifications were performed using the appropriate software. All statistical tests were performed using Excel 2016 or Prism 10. The statistical analyses performed for each experiment are indicated in figure legends.

## Reporting summary

Further information on research design is available in the Nature Portfolio Reporting Summary linked to this article.

# Data availability

RNA sequencing data using B16F10 cells are available under the accession number GSE286709. Quantitative proteomics data on protein aggregates are available on PRIDE under the number PXD059835 (https://www.ebi.ac.uk/pride/). Spatial transcriptomics dataset is available under the accession number GSE316760. All other sequencing data are deposited on GEO under the accession number GSE286704 for the ribosome profiling and GSE286715 for the RNAseq after Flag-RIG-IP (https://www.ncbi.nlm.nih.gov/geo/query/acc.cgi? acc=GSE286715). Results are in part based on TCGA-SKCM data generated by the TCGA Research Network (http://cancergenome.nih.gov/). Source data are provided as a Source Data file. All other data supporting the findings of this study are available from the corresponding author on reasonable request (Pierre.Close@uliege.be). Source data are provided with this paper.

# Code availability

TrimGalore: https://www.bioinformatics.babraham.ac.uk/projects/trim_galore/ TrimGalore is a Perl wrapper based on two canonical tools (Cutadapt and FastQC) for adapters and quality-based trimming. TrimGalore's usage improves consistency and repeatability of analysis. No newly generated code is presented. RSamtools: https://bioconductor.org/packages/release/bioc/html/Rsamtools.html. R package of Samtools was used for BAM files manipulations in this project (i.e: Metagene plot). No newly generated code is presented. GenomicAlignments: https://bioconductor.org/packages/release/bioc/html/GenomicAlignments.html R/Bioconductor package was used for the storage, manipulation and representation of short genomic alignments. In this project this package allowed us to perform the binning of genomic alignments for the ribosome stalling analysis. No newly generated code is presented. Pysam: https://github.com/pysam-developers/pysam: this used for the manipulation and storage of BAM/SAM files. Here, it was used to store BAM files for the tRNA pools analysis. No newly generated code is presented. ToppGene: https://toppgene.cchmc.org/: ToppGene suite is a canonical tool to perform Gene Ontology analysis, gene set enrichment and candidate gene

prioritization. It was used for the RNA-sequencing data analysis. No newly generated code is presented. STAR: https://github.com/alexdobin/STAR: STAR is a canonical software for mapping RNA-seq reads to a reference genome. It was used for the RNA-seq and ribosome profiling in this project. No newly generated code is presented. DESeq2: https://bioconductor.org/packages/release/bioc/html/DESeq2.html: it was used for the normalization, visualization, analysis of RNA-seq data and differential expression analysis. Used for the RNA-sequencing and ribosome profiling analysis. No newly generated code is presented. EdgeR https://bioconductor.org/packages/release/bioc/html/edgeR.html: R package developed for the differential expression analysis of RNA-sequencing data. It was used here for statistical analysis in the ribosome profiling analysis. No newly generated code is presented. GSVA: https://www.bioconductor.org/packages/release/bioc/html/GSVA.html: Gene Set Variation Analysis R package allowing gene set and pathway enrichment on single samples. No newly generated code is presented. Source data files are provided with this paper in supplementary data files. Gating strategies for FACS analyses is available in the corresponding supplementary figures.

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

## Acknowledgements

We thank Ghanem Ghanem and Ahmad Najem for the access to MM lines, and Lionel Larue for the access to M1014 cells. We are grateful to the GIGA-proteomics, imaging, genomics, bio-informatics and viral vector facilities for their assistance. This study was supported by the Belgian foundation against Cancer (2020-068; 2024-148), the Walloon Excellence in Life Sciences and Biotechnology (WEL Research Institute, WELBIO to P. Close), the FNRS (PDR T.0244.18; EOS O.0020.22), the University of Liege, and the "Foundation Leon Fredericq". CD was supported by a FNRS Télévie grant. SD was supported by a FNRS research fellow grant, by a Cancer Research UK institute award (C5759/27412) and a Royal Society research grant (RGS/R2/252386), AB and FR are Research Associates, AC and PC are Research Directors at the FNRS, respectively. TB and AV were supported by the National Institute for Health and Care Research (NIHR) Manchester Biomedical Research Centre (BRC) (NIHR203308), Cancer Research UK RCCASF-May23/100001 Cancer Research UK Advanced Clinician Scientist, a core funded grant to the Cancer Research UK Manchester Institute (C5759/A27412), Melanoma Research Alliance and Rosetrees Trust Young Investigator Award (#825648). The views expressed are those of the author(s) and not necessarily those of the NIHR or the Department of Health and Social Care. This article is based upon work from COST Action TRANSLACORE CA21154, supported by COST (European Cooperation in Science and Technology).

## Author contributions

C.D. designed, performed and analysed in vitro and in vivo experiments, and wrote the manuscript. C.S. assisted with orthotopic transplantation assays, FACS and analyses. C.C. and F.R. performed computational analyses of the RNA sequencing, Ribosome profiling, Proteomics and RNA immunoprecipitation sequencing data. C.M. and A.C. performed and analysed histological staining of tumour samples. A.B., N.A., D.H., R.V. performed immunoblotting experiments. N.E.-H. performed polysome sequencing. T.B. and A.V. generated and analysed the spatial transcriptomics dataset. L.M.-M. and F.R. performed ribosome profiling experiments. E.R. supported in vivo experiments. M.S.R. performed Nicoletti assays. M.L. performed computational analyses using TCGA

patient data. J.U. and A.V. provided expertise for human clinical data. S.D. and P.C. supervised the work, designed and analysed experiments, and wrote the manuscript. P.C. acquired and secured funding. All authors discussed the results and commented the manuscript.

## Competing interests

P.C. and F.R. are co-founders and scientific advisors at THERAtRAME SA. All other authors declare no conflict of interest.

## Additional information

[1]Laboratory of Cancer Signaling, GIGA-Institute, University of Liège, Liège, Belgium. [2]Laboratory of Cancer Stemness, GIGA-Institute, University of Liège, Liège, Belgium. [3]Laboratory of Cancer biology, GIGA-Institute, University of Liège, Liège, Belgium. [4]Laboratory of Metabolic Regulation, GIGA-Institute, University of Liège, Liège, Belgium. [5]Skin Cancer and Ageing Lab, Cancer Research UK Manchester Institute, The University of Manchester, Manchester, UK. [6]Department of Dermatology, Salford Royal NHS Foundation Trust, NIHR Manchester Biomedical Research Centre, The University of Manchester, Manchester, UK. [7]Department of Molecular and Clinical Cancer Medicine, The University of Liverpool, Liverpool, UK. [8]WELBIO department, WEL Research Institute, Wavre, Belgium. [9]Skin Cancer Unit, German Cancer Research Center (DKFZ), Heidelberg, Germany. [10]Department of Dermatology, Venereology and Allergology, University Medical Centre Mannheim, Ruprecht-Karl University of Heidelberg, Mannheim, Germany. [11]DKFZ Hector Cancer Institute at the University Medical Centre Mannheim, Germany Cancer Research Centre – Deutsches Krebsforschungszentrum (DKFZ), Mannheim, Heidelberg, Germany. [12]Laboratory of RNA dynamics in Cancer, Cancer Research UK Manchester Institute, The University of Manchester, Manchester, UK. [13]These authors contributed equally: Christian Seca, Coralie Capron. [14]These authors jointly supervised this work: Sylvain Delaunay, Pierre Close. ✉e-mail: Pierre.Close@uliege.be

