## [Transparent Peer Review file · Nature Communications]

Disruption of tRNA threonylation triggers RIG-I mediated anti-tumour immune response

Corresponding Author: Dr Pierre Close

Version 0:

Reviewer comments:

Reviewer #1

(Remarks to the Author)

In this manuscript, the authors identify the role of OSGEP inactivation in promoting T-cell infiltration in vivo in melanoma cells. Mechanistically, they show that OSGEP function is required to prevent the formation of misfolded protein granules due to translation defects. In turn, these trigger the formation of cytoplasmic stress granules and the release of pro-inflammatory cytokines through the activity of HRI and RIG-1, respectively. While the effect of OSGEP depletion on protein homeostasis was previously described, its effect on T-cell activation and the anti-tumoral response is novel and highly interesting. Despite this, certain aspects of the study remain unclear. The authors should address these before the manuscript can be considered for publication.

Major points:

1. In Figure 2, the authors identify a gene expression signature mediated by OSGEP downregulation in melanoma cells. They focused on upregulated transcripts, showing that the same upregulation correlates with good prognosis in melanoma patients. Did the authors investigate whether there is any negative correlation between this signature and OSGEP expression?
2. Since t6A on tRNA can be detected through mass spectrometry, the authors should investigate the abundance of this modification in primary melanoma samples and untransformed nevi or normal skin tissue, and its correlation with OSGEP expression and prognosis.
3. In Figure 4, the authors show that OSGEP inactivation, while it doesn't affect global translation levels, increases ribosome occupancy on ANN codons of mRNA encoding for proteins found in protein aggregates. It is not clear how the translation of a relatively small number of mRNAs is affected by OSGEP depletion. The authors should analyze the ribosome footprinting dataset to identify potential ribosome stalling or frameshift specifically occurring on affected mRNAs. Are total levels of misfolded proteins affected by OSGEP depletion?
4. Connected to my previous point, in Figure 2K, it is shown that increased occupancy only occurs on specific ANN codons. The authors should report occupancy for individual codons in Figures 4e, h, i, and j. How can the authors justify this observation?
5. In order to better characterize their mechanism, the authors should perform rescue experiments by transfecting modified or unmodified ANN tRNAs in OSGEP-depleted cells and investigate protein granule formation and downstream activation of HRI and RIG-1.
6. In Figure 6, it is reported that OSGEP depletion increases RIG-1 interaction with tRNAs. Again, the authors did not observe any specificity for OSGEP tRNA targets. Did they analyze specific enrichment of individual OSGEP targets?
7. Is the presence of unmodified NNA tRNAs sufficient for RIG-1 activation? To test this, the authors should transfect unmodified NNA tRNAs in melanoma cells and investigate RIG-1 activation.

Minor points:

1. In Figure 1, the levels of Ki67 and Caspase 3 in tumors grown in immune-deficient mice should be reported.
2. In Figure 3F, the immunoblotting analysis of M1014 cells shows differences in HSC70 normalization. The authors should include a better-normalized Western blot or perform densitometric quantification of the current one.

Reviewer #2

(Remarks to the Author)
Comments to the Author
Summary/General Comments:

This study presents a timely and relevant research topic, reveals the key role of OSGEP-mediated t6A tRNA modification in melanoma immune evasion. The authors elegantly demonstrate that loss of OSGEP disrupts protein translation, triggers stress granule formation, and activates RIG-I signaling, ultimately driving T-cell recruitment and tumour regression. Additionally, the identification of an OSGEP-related gene signature correlates with increased T-cell infiltration and better survival, highlighting t6A modification as a potential therapeutic target in melanoma.

Major/minor Comments:

1. Overall, the manuscript is well written, and the study presents novel and impactful biological insights with clear clinical relevance.
2. The transcriptomics data analysis is comprehensive and well-integrated with the experimental findings. However, I wonder about the rationale for selecting only the top 100 upregulated genes following OSGEP depletion in B16F10 cells. Would the main findings and biological interpretation remain consistent if different thresholds (e.g., top 10, 20, 50, or even 500 genes) were applied? It would be helpful if the authors can clarify on the robustness of their results to this parameter selection.
3. Some minor issues need clarification:
 - In Figure 1, the term esiRNA appears — can the authors clarify or briefly explain this term upon its first usage?
 - There is a general lack of abbreviation definitions throughout the manuscript. Please ensure all abbreviations are defined at first mention for clarity.
4. Regarding the statistical analysis, it would be more appropriate to report adjusted p-values (FDR-adjusted p-values) rather than exact p-values alone, particularly in large-scale transcriptomics analyses, to better control for false discovery rate (FDR) and reduce potential false positives.

Reviewer #3

(Remarks to the Author)

This manuscript presents a novel and compelling mechanistic link between tRNA modifications, proteostasis, and immune surveillance in cancer.

Carefully executed study that identifies a novel role for the t6A tRNA modification enzyme OSGEP in immune evasion and tumor progression. The authors convincingly demonstrate that loss of OSGEP induces proteostasis defects, leading to RIG-I activation and a CD4+/CD8+ T cell-mediated anti-tumor response in melanoma. The mechanistic insights are supported by in vitro, in vivo, and transcriptomic data, and the clinical relevance is validated using patient-derived datasets. The manuscript is overall clear and comprehensive. The figures are well-organized, and the data supports the authors' conclusions. We believe this work will be a valuable contribution to the community. However, before publication, we encourage the authors to carefully address both the major and minor comments provided.

Major comments

1. Are other members of the KEOPS complex involved in this phenotype?
 - The manuscript focuses exclusively on OSGEP, but does not assess whether other KEOPS subunits (e.g., TP53RK, TPRKB, LAGE3, GON7) contribute to the observed immune activation.
 - Would an OSGEP IP help clarify whether it acts independently or as part of the KEOPS complex in this context?
2. Figure 3e, f; 5a, b. Have the authors tested human melanoma cell lines?
 - Cell lines could increase the translational relevance of the findings.

3. The study address that RIG-I activation upon OSGEP loss is mediated by tRNA accumulation. However, given that silencing HRI or ISRIB treatment suppresses RIG-I activation, could the authors clarify whether tRNA detection is sufficient on its own, or whether additional cytoplasmic events (e.g., stress granule formation, aggregation) are required to make tRNAs immunogenic?

Minor comments:

-Codon occupancy analysis (Figure 4e): The legend of Fig. 4e refers to codon occupancy in general terms, but the main text specifies that the analysis focuses on the A-site. Since ribosome profiling allows analysis at different ribosomal sites (A, P, E), it is important to clearly indicate in the figure legend that codon occupancy was measured at the A-site.

-RIG-I RIP-seq (Fig. 6a-c): the number of biological/technical replicates used for sequencing is not specified.

-While the authors state that ribosome occupancy on ANN codons is increased in OSGEP-deficient cells, especially in mRNAs encoding aggregated proteins, Extended Fig. 8e does not show a clear or statistically significant increase in A-site occupancy on ANN codons.

-In Extended Figure 6c, the heatmap suggests there are changes in some tRNA expression between shCTRL and shOSGEP. This appears inconsistent with the statement in lines 213–215 ('Of note, the lack of t6A modification did not lead to changes in the tRNA pool in B16F10 cells, showing the deposition of t6A on tRNAs NNU does not affect their stability') that the tRNA pool remains unchanged and stable. The authors should clarify whether the changes shown in Fig. 6c are statistically or biologically significant, and how they reconcile this with their claim that t6A does not affect tRNA stability.

-In Figure 6d, the authors state that tRNAs from terminal base anticodon groups (NNU, NNA, NNC, NNG) were "uniformly upregulated." However, the figure only shows the relative proportions of these groups among RIG-I-enriched tRNAs, without comparison to the total tRNA pool or to non-enriched species. The term "uniformly upregulated" may be misleading and should be clarified.

Reviewer #4

(Remarks to the Author)

In their study "Disruption of tRNA threonylation triggers RIG-I mediated antitumour immune response" Dziagwa et al. observe that knock-down of O-sialoglycoprotein endopeptidase (OSGEP) which catalyses the tRNA modification N6-threonylcarbamoyladenosine (t6A) leads to RIG-I activation in melanoma cells.

The authors performed an RNAi screen against tRNA modifying enzymes in patient-derived melanoma cell lines (i.e. MM117, MM011) and monitored protein aggregates as a read-out of mistranslation. Here, repression of OSGEP caused the strongest mistranslation response. This finding was validated by shRNAs in MM117, MM011 and in murine melanoma cell lines M1014, B16F10. OSGEP is responsible for the modification t6A in position 37 of the cytoplasmic tRNAs "tRNANNU". Repression of OSGEP reduced tumor size in immune competent but not in immune deficient (NOD-SCID) mice. While B16F10 tumours were poorly infiltrated, OSGEP deficient tumours had a robust infiltration of T cells, partially by NK cells, but not B cells, macrophages or neutrophils. T cell but not NK cell depletion abolished the tumour growth defect of OSGEP knockdown tumours. Upregulation of genes induced by OSGEP deficiency correlated with tumor survival in SKCM patients. Knockdown of OSGEP enhanced migration of naïve T cells ex-vivo. Further data reveal that knockdown of OSGEP upregulates RIG-I expression and RIG-I signaling/activation. Interestingly, the depletion of OSGEP did not impair global protein synthesis as monitored by polysome profiles. However, RNAi of t6A affected the decoding of ANN codons at the A-site and the integrated stress response pathway was induced via HRI upregulation and eIF2a phosphorylation. The authors also observed colocalization of RIG-I with G3BP1 in cytosolic stress granules upon RNAi of OSGEP.

The depletion of HRI abolished the formation of stress granules observed in OSGEP-deficient cells (Fig. 5f-g), and also prevented RIG-I activation in OSGEP-deficient cells. By contrast pharmacologic inhibition of stress granule formation did not prevent the formation of protein aggregates.

Knock-down of RIG-I in OSGEP deficient cells rescued immune evasion and prevented T cell infiltration in B16F10 tumours. RIG-I/RNA-complex IP revealed enrichment of tRNA binding to RIG-I upon OSGEP depletion. However, not the N6-threonylcarbamoyladenosine containing tRNAs were enriched in the RIG-I binding RNA pool, indicating that RIG-I recognizes endogenous tRNA independently of t6A modification in stress granules upon HRI activation.

Major comments

This is a very interesting study. It would be crucial to know how RIG-I activation works.

1. However, it remains unclear whether a tRNA modification dependent stress response that activates RIG-I is a result of missing RNA modifications leading to RIG-I recognition or if the conditions of stress lead to a different RIG-I activation by endogenous RNA or independently of any RNA recognition. Since binding to RIG-I is necessary but not sufficient for RIG-I activation the changed binding of tRNAs to RIG-I is only a hint but not a proof for RIG-I activation by tRNA and it is unexpected that RIG-I would recognize endogenous tRNAs. The question is whether OSGEP depletion induce the generation of RIG-I ligands or if HRI activity in all cases leads to recognition of endogenous RNA by RIG-I and that enrichment of RIG-I binding tRNA represents an incidence.

a) The authors should induce HRI activation by other mechanisms (e.g. by proteasome inhibitors, bortezomib or MG132, PMID: 21090173) and check if HRI activation in general leads to higher tRNA binding to RIG-I in B16F10 melanoma cells or if this effect is specific for OSGEP depletion.

b) The authors should purify the RNA bound to RIG-I and total RNA from OSGEP depleted cells and WT cells and stimulate WT cells with these RNAs. If OSGEP deficiency produces RIG-I ligands, the RNA of OSGEP deficient cells but not of ctrl shRNA cells should activate RIG-I in WT cells. I

2. eIF2alpha phosphorylation should cause global translational inhibition. How can a polysome profile be unaffected if eIF2a is phosphorylated?

The authors should use another read-out to confirm that global protein synthesis is not influenced (e.g. SUnSET assay).

3. Is it excluded that OSGEP is required for other modifications than t6A?

Minor comments

Why does RIG-I activation not induce cell death/growth reduction by cell autonomous mechanisms.

Reviewer #5

(Remarks to the Author)

The manuscript entitled "Disruption of tRNA threonylation triggers RIG-I mediated anti-tumour immune response" presents an intriguing study exploring how alterations in tRNA modifications may influence protein aggregation, leading to activation of the RIG-I pathway and subsequent induction of type I interferon responses that enhance antitumor immunity. The authors provide a coherent set of experiments supporting the central hypothesis and demonstrate mechanistic links between tRNA modification, stress responses, and immune activation. However, there are several major concerns that limit the impact and novelty of the study.

1. Limited Cell Line Validation:

The mouse model findings are derived primarily from a single cell line, which substantially weakens the generalizability of the conclusions. Validation across additional cell types would be necessary to support the broader relevance of the proposed mechanism.

2. Neglect of Potential Neoantigenic Consequences:

Alterations in tRNA modifications are likely to influence translational fidelity and could result in neoantigen generation. This possibility is not explored, nor is its potential contribution to antitumor immune responses addressed. This represents a significant gap in the mechanistic understanding of how tRNA modifications may affect immune surveillance.

3. Lack of Novelty in RIG-I/Type I Interferon Axis:

Although the mechanistic connection between protein aggregation and RIG-I activation is of interest, the involvement of the RIG-I/type I interferon pathway in antitumor immunity has been reported in multiple previous studies. Without a compelling new insight or clinical relevance, the novelty of this work remains limited.

Version 1:

Reviewer comments:

Reviewer #1

(Remarks to the Author)

The authors addressed my major and minor points, the manuscript can be now considered for publication.

Reviewer #2

(Remarks to the Author)

Reviewer Report

The authors have responded to reviewer feedback; however, concerns remain regarding transparency and consistency between the rebuttal letter and the revised manuscript.

A) While the authors assert in the rebuttal letter that the concerns have been addressed and the relevant sections updated, these changes are not visible in the Figure 2 legend or the main text. This inconsistency calls into question the reliability of the reported revisions.

#The transcriptomics data analysis is comprehensive and well-integrated with the experimental findings. However, I wonder about the rationale for selecting only the top 100 upregulated genes following OSGEP depletion in B16F10 cells. Would the main findings and biological interpretation remain consistent if different thresholds (e.g., top 10, 20, 50, or even 500 genes) were applied? It would be helpful if the authors can clarify on the robustness of their results to this parameter selection.

We apologize for the initial wording ambiguity. Rather than using a fixed "top 100 transcripts UP" cutoff, our analysis applied thresholds of fold change (FC) > 0.5 and p-value < 0.05, which yielded approximately ~120 upregulated genes. Among these, 94 were matched in the TCGA dataset, forming the final OSGEP gene set. To improve transparency, we have updated Figure 2h–i by removing the "top 100 transcripts UP" label and replacing it with the precise criteria: "FC > 0.5, p < 0.05". The methods and figure legends have been updated accordingly. We thank the reviewer for highlighting this, which allowed us to strengthen the rigor and clarity of our analysis.

h. Volcano plot of the expression of transcripts 656 in shCtr versus shOSGEP B16F10 cells and generation the OSGEP gene set signature 657 consisting of the top 100 genes upregulated. i. Heatmap of unsupervised clustering 658 identified 2 clusters of patients with SKCM (TCGA) according to the top 100 upregulated 659 genes from the OSGEP gene set signature

(cluster 1, n = 251; cluster 2, n = 222)

B) The authors have addressed the minor issues raised previously.

Reviewer #3

(Remarks to the Author)

The authors have fully addressed all our previous concerns with appropriate new experiments, improved statistical analysis, and clearer figure legends. The mechanistic model connecting t6A deficiency to stress-granule-dependent RIG-I activation is now much more convincing. The addition of LAGE3 knockdown, human melanoma validation and RNA immunogenicity tests significantly strengthen the manuscript's translational relevance.

We recommend only minor additional clarifications:

1. Regarding sex reporting in animal experiments, Nature Communications requires explicit disclosure of the sex of the mice used, even if no sex-based analysis was performed. In the "Methods - Animal models" section (currently stating only the strain and age), the authors should indicate whether C57BL/6 and NOD-SCID mice were males, females, or mixed, and justify the absence of sex-stratified analyses in accordance with SAGER guidelines.

2. A brief explanatory note in the Results accompanying Fig. 4c/d and Extended Data Fig. 9 would help clarify why eIF2 α phosphorylation despite the absence of measurable global translational repression, noting that the ISR induction observed is modest and selective.

Reviewer #4

(Remarks to the Author)

The manuscript improved a lot! Minor comment: It would be great if the read-out of a stimulation with RIG-I ligands from a RIG-I pull-down would not be RIG-I expression (Figure 6a.). Since RIG-I is an ISG I believe that the conclusions are correct, however this read-out would confuse the reader and one could speculate that the ligand influences RIG-I protein stability. Better show IFIT, ISG15, STAT1/2 phosphorylation or MX protein western blot as a read-out of type I IFN induction.

Reviewer #5

(Remarks to the Author)

While the novelty of the work remains somewhat limited, the authors have sufficiently addressed all other concerns raised in the previous review.

Reviewer #6

(Remarks to the Author)

Manuscript
NCOMMS-25-15763-T

We would like to sincerely thank the reviewers for their careful evaluation of our work and their constructive comments. We are very pleased that the novelty and solidity of our study were recognized, and we greatly appreciate the insightful suggestions that helped us further really improve the manuscript.

We systematically **addressed all major and minor points raised**, either by performing new experiments or by providing additional analyses and clarifications. The revised version of the manuscript now includes a substantial amount of new data and figures that refine the mechanistic understanding of how OSGEP regulates tumor immunity and further support the translational relevance of our findings.

In summary, we provide the following additional evidence:

- 1) **Involvement of t6A tRNA modification** in the OSGEP phenotypes: We performed additional experiments depleting LAGE3, another essential KEOPS subunit, confirming that the OSGEP phenotypes result from the loss of t6A tRNA modification rather than unrelated effects.
- 2) **Ribosome stalling**. Using RiboDiPA pipeline, we demonstrate that OSGEP depletion induces ribosome stalling on mRNAs encoding proteins accumulating in aggregates.
- 3) We show that small RNAs generated upon OSGEP depletion are sufficient to activate RIG-I in wild-type cells, highlighting their **immunogenic potential**.
- 4) **Human cell line and additional model**: We validated our main findings in human melanoma cells and in an independent melanoma mouse model.
- 5) **Transcriptomic data**: we strengthened the analysis and description of the transcriptomic data. In addition, we now provide spatial transcriptomic data from melanoma patient samples, supporting translation relevance of our observations.
- 6) We have clarified the methods and conclusions across the paper and performed all requested experiments to ensure robust validation of our conclusions.

Please find below our detailed point-by-point responses to the reviewer's questions.

REVIEWER COMMENTS

Reviewer #1:

In this manuscript, the authors identify the role of OSGEP inactivation in promoting T-cell infiltration in vivo in melanoma cells. Mechanistically, they show that OSGEP function is required to prevent the formation of misfolded protein granules due to translation defects. In turn, these trigger the formation of cytoplasmic stress granules and the release of pro-inflammatory cytokines through the activity of HRI and RIG-1, respectively. While the effect of OSGEP depletion on protein homeostasis was previously described, its effect on T-cell activation and the anti-tumoral response is novel and highly interesting. Despite this, certain aspects of the study remain unclear. The authors should address these before the manuscript can be considered for publication.

We sincerely thank the reviewer for the constructive and encouraging comments on our work. We are pleased that the novelty of our findings regarding the role of OSGEP inactivation in promoting T cell infiltration and antitumor immunity has been recognized. In the revised manuscript, we have addressed in detail the seven major points and the minor comments raised by the reviewer. These revisions clarify the molecular mechanisms underlying OSGEP depletion-induced protein misfolding, stress granule formation, and the consequent activation of HRI-RIG-I signaling leading to cytokine release and enhanced T-cell recruitment.

Major points:

1. In Figure 2, the authors identify a gene expression signature mediated by OSGEP downregulation in melanoma cells. They focused on upregulated transcripts, showing that the same upregulation correlates with good prognosis in melanoma patients. Did the authors investigate whether there is any negative correlation between this signature and OSGEP expression?

We thank the referee for this insightful comment. We investigated whether OSGEP expression negatively correlates with the level of the identified OSGEP gene set in melanoma patient datasets and did not observe a strong inverse correlation (**now shown in *Extended Data Figure 6b***). We note, however, that OSGEP expression remains rather homogeneously expressed in melanoma tumors. In our experimental, we depleted OSGEP using shRNA to study its mechanistic consequences on tumor immunogenicity, leading to decreased OSGEP expression beyond the expected physio-pathological conditions.

Importantly, our functional data demonstrate that experimental targeting of OSGEP triggers the upregulation of a distinct immunogenic gene expression program, which we show is associated with improved prognosis in melanoma patients. This supports the notion that OSGEP functions as a gatekeeper of immune evasion and suggests that therapeutic inhibition of OSGEP could reprogram the tumor microenvironment to favor anti-tumor immunity. Therefore, even if inverse correlation is not evident at the bulk RNA level in patient's datasets, the pathway remains clinically relevant.

2. Since t6A on tRNA can be detected through mass spectrometry, the authors should investigate the abundance of this modification in primary melanoma samples and untransformed nevi or normal skin tissue, and its correlation with OSGEP expression and prognosis.

We thank the reviewer for this constructive suggestion. We explored the possibility of assessing t⁶A abundance in patient material. However, technical limitations currently preclude reliable quantification of this modification in small amounts of RNA obtained from clinical biopsies. Instead, we evaluated OSGEP expression in primary melanoma tumors compared to adjacent normal skin tissue. Our data show that OSGEP expression is significantly upregulated in tumors (**now shown in the revised manuscript in *Extended Data Figure 5f-g***). We also analyzed *The Cancer Genome Atlas* (TCGA) for patients with *Skin Cutaneous Melanoma* (SKCM), and did not detect a significant correlation between OSGEP mRNA expression and overall survival in melanoma patients.

This result is consistent with our central hypothesis: the downregulation of OSGEP in our study represents an experimental perturbation (via shRNA-mediated depletion) designed to uncover mechanistic consequences on tumor immunogenicity, rather than a physiological event in patient tumors. Given that loss of OSGEP function impairs t⁶A modification and leads to the activation of innate immune sensing pathways, we propose that such downregulation does not occur spontaneously in patient tumors but can be leveraged to unlock an immunogenic state. Importantly, our data demonstrate that OSGEP depletion induces an immunogenic gene expression program (OSGEP gene set) that is positively associated with favorable prognosis in melanoma (*Figure 2h-m*).

Together, these findings support the clinical relevance of OSGEP as a therapeutic target, while highlighting the distinction between physiological expression levels in tumors and the mechanistic consequences of its targeted inhibition.

3. In Figure 4, the authors show that OSGEP inactivation, while it doesn't affect global translation levels, increases ribosome occupancy on ANN codons of mRNA encoding for proteins found in protein aggregates. It is not clear how the translation of a relatively small number of mRNAs is affected by OSGEP depletion. The authors should analyze the ribosome footprinting dataset to identify potential ribosome stalling or frameshift specifically occurring on affected mRNAs. Are total levels of misfolded proteins affected by OSGEP depletion?

We thank the reviewer for this insightful comment. Our ribosome profiling data show increased ribosome occupancy on ANN codons in mRNAs encoding proteins found in aggregates from shOSGEP cells. To answer the reviewer's comment, we analyzed ribosome stalling using **RiboDipa** (*Li et al, Nucleic Acids Research, 2020*). We calculated the average ribosome stalling (RPF count/gene) for each mRNA encoding proteins found in aggregates in control and OSGEP knockdown cells, expressed as log₂FC (shOSGEP/ shCtr). This analysis revealed a significant increase in ribosome stalling values for OSGEP-depleted aggregate mRNAs compared to controls (Rev1 Fig. 1a, Mann Whitney test). Strikingly, this effect correlated with increased GCN2 activation, a known sensor of ribosome stalling, in OSGEP-depleted cells (Rev1 Fig. 1b). Together, these new data demonstrate that OSGEP (t⁶A) depletion promotes specific ribosome stalling on mRNAs of proteins found in aggregates.

Finally, previous studies in yeast and *Drosophila* have reported that loss of t⁶A alters translational fidelity and promote protein misfolding (*Pollo-Oliveira et al, Biomolecules 2020; Rojas-Benitez et al, Biomolecules 2017; Thiaville et al., Microbial Cell 2015*), consistent with our observation that OSGEP depletion increases protein aggregation in mammalian cells. We thank the reviewer for the suggestion and have now included the new panel ***in the revised manuscript in Figure 4h***.

Reviewer 1 Figure 1: a. Ribosome stalling analysis performed with RiboDipa. Ribosome stalling (RPFs count per gene) for each mRNA encoding proteins found in aggregates in shCtr and shOSGEP cells, expressed as log₂FC (shOSGEP/ shCtr) was calculated. pvalue was calculated by Mann-Whitney test. **b.** Western blot analysis of indicated proteins from B16F10 cells depleted or not for OSGEP.

4. Connected to my previous point, in Figure 2 4K, it is shown that increased occupancy only occurs on specific ANN codons. The authors should report occupancy for individual codons in Figures 4e, h, i, and j. How can the authors justify this observation?

We thank the reviewer for this helpful suggestion. Codon occupancy for each individual codon is **now provided in Extended Data Figure 9e** (related to main figure 4e), which shows the analysis across all ribosome-protected fragments. In addition, we included codon occupancy analyses for transcripts corresponding to proteins found in shOSGEP aggregates, shCtr aggregates, as well as a random set of genes (**newly added Extended Data Figure 10a**). These results confirm that increased ribosome occupancy is enriched at specific ANN codons, consistent with the requirement of OSGEP-dependent t⁶A modification for efficient decoding of ANN triplets. Thus, the observed codon-specific effect reflects the direct mechanistic consequence of OSGEP inactivation on translation.

5. In order to better characterize their mechanism, the authors should perform rescue experiments by transfecting modified or unmodified ANN tRNAs in OSGEP-depleted cells and investigate protein granule formation and downstream activation of HRI and RIG-1.

We thank the reviewer for this interesting comment. To address this point, we performed rescue experiments by overexpressed four tRNAs target of OSGEP together in control or OSGEP depleted cells. We focused on the tRNA corresponding to the codons for which differences in ribosome occupancy were observed in original Figure 4k - new Figure 4l: tRNA^{ASN}_{AUU} (codon AAU), tRNA^{LYS}_{UUU} (codon AAA), tRNA^{ILE}_{UAU} (codon AUA) and tRNA^{LYS}_{CUU} (codon AAG). Our results show that tRNA overexpression alone does not lead to RIG-I activation in control cells. As expected, OSGEP depletion induced RIG-I activation, but importantly, tRNA overexpression did not attenuate this response, indicating that supplying additional ANN tRNAs is not sufficient to rescue the phenotype (Rev1 Fig. 2).

Strikingly, this finding is consistent with observations in yeast reported by *Thiaville et al (Microbial Cell, 2015)*, where t⁶A-deficient phenotypes could not be suppressed by expressing ANN-decoding tRNAs.

Reviewer 1 Figure 2: Western blot analysis of RIG-I, OSGEP and GAPDH (loading control) of control or OSGEP-depleted B16F10 cells, overexpressing or not the four tRNA: tRNA^{ASN}_{AAU} (codon AAU), tRNA^{LYS}_{UUU} (codon AAA), tRNA^{ILE}_{UAU} (codon AUA) and tRNA^{LYS}_{CUU} (codon AAG).

6. In Figure 6, it is reported that OSGEP depletion increases RIG-I interaction with tRNAs. Again, the authors did not observe any specificity for OSGEP tRNA targets. Did they analyze specific enrichment of individual OSGEP targets?

We fully agree with the reviewer's observation. Since RIG-I is a well-established RNA sensor and OSGEP depletion activates an immune response via RIG-I, we investigated the nature of the RNA ligands involved. To this end, we performed RNA immunoprecipitation followed by sequencing of FLAG-tagged RIG-I in OSGEP-depleted cells (*Figure 6a*). This analysis revealed a marked increase in RIG-I association with tRNAs (*Figure 6b-c*).

As noted by the reviewer, while RIG-I binding to tRNAs was broadly enhanced, we did not observe preferential enrichment of individual OSGEP modified tRNAs. Instead, RIG-I interacted with multiple anticodon families (NNU, NNA, NNC, NNG), suggesting a general increase in tRNA recognition rather than selective binding to t⁶A-dependent targets (*Figure 6d* and *Extended Data Figure 13b-d*).

Together, these findings support a model in which OSGEP depletion leads to widespread tRNA recognition by RIG-I, possibly due to altered cellular context, protein misfolding stress and stress granule localization, rather than selective enrichment of OSGEP-specific tRNA substrates.

7. Is the presence of unmodified NNA tRNAs sufficient for RIG-I activation? To test this, the authors should transfect unmodified NNA tRNAs in melanoma cells and investigate RIG-I activation.

We thank the reviewer for this important point. Given that OSGEP specifically modifies ANN-decoding tRNAs via the t⁶A modification, we reasoned that the main difference between the small RNA pools derived from shCtr and shOSGEP cells is most likely attributable to tRNAs and their modification status. While we initially hypothesized that hypomodified tRNAs would be preferentially recognized by RIG-I, our RIP-seq analyses revealed that RIG-I associates broadly with multiple tRNA species, independent of their t⁶A modification status.

To directly assess the immunogenicity of these RNAs, we extracted small RNAs from shCtr and shOSGEP B16F10 cells and transfected them into wild-type (WT) B16F10 cells.

Importantly, only small RNAs from OSGEP-depleted cells induced RIG-I activation. These new results have been included **in the revised manuscript in Figure 6a**. To further dissect the relationship between stress granules and activation of RIG-I by tRNAs, we treated B16F10 cells with sodium arsenite (a stress granule inducer; see for example *McEwen et al, J Biol Chem, 2005; Ziqi Ren et al, Nat Commun, 2023; Ho Chun Lai et al, Mol Biol Cell 2024*) and/or with small RNAs extracted from shCtr or shOSGEP cells (Rev1 Fig. 3). Notably, sodium arsenite-induced stress granules and small RNAs from control cells (shCtr) were insufficient to activate RIG-I. However, treatment with small RNAs from OSGEP-depleted cells, either alone or with sodium arsenite, significantly increased RIG-I levels. In fact, treatment of B16F10 cells with small RNA from OSGEP-depleted cells already induced stress granule formation, without the need to induce them with sodium arsenite (Rev3 Fig.1). Together, these findings clarify that both OSGEP-dependent small RNAs and stress granule formation are required for RIG-I activation, highlighting a sequential mechanism in which OSGEP loss creates the cellular context prone to induce an anti-tumor immune response.

Reviewer 1 Figure 3: a. Western blot of indicated proteins from B16F10 cell infected by small RNA extracted with mirvana kit from control or OSGEP cells. Cells have been then treated with DMSO or Arsenite (200µM for 2h). **b.** Representative immunofluorescence images of B16F10 cells cell infected by small RNA extracted with mirvana kit from Control or OSGEP cells. Cells have been then treated with DMSO or Arsenite (200µM for 2h). Stress granules (staining G3BP1) are shown in green and nuclei (DAPI) in blue. **c.** Quantification of stress granules in B16F10 cells cell infected by small RNA extracted with mirvana kit from control or OSGEP cells, treated or not with Arsenite. Data are reported as mean +/- SD. Exact p-values are indicated.

Minor points:

1. In Figure 1, the levels of Ki67 and Caspase 3 in tumors grown in immune-deficient mice should be reported.

We thank the reviewer for this suggestion. To address this point, we performed immunohistochemical staining for Ki67 and cleaved Caspase-3 in B16F10 tumors derived from NOD-SCID mice. Consistent with the absence of an effect on tumor growth in immunodeficient hosts (*Fig. 1f-g; Extended Data Fig. 1e-g*), we did not observe any significant difference in proliferation or apoptosis between control (shCtr) and OSGEP-depleted (shOSGEP#2) tumors. These new data are now included **in the revised manuscript as Extended Data Figure 2g-h**.

2. In Figure 3F, the immunoblotting analysis of M1014 cells shows differences in HSC70 normalization. The authors should include a better-normalized Western blot or perform densitometric quantification of the current one.

We performed a densitometric quantification pTBK1 bands across three independent replicates, normalizing to HSC70. This analysis confirmed that OSGEP knockdown in both B16F10 and M1014 cells leads to an approximately twofold increase in pTBK1 levels. We have now included these quantification panels **in the revised manuscript as *Extended Data Figure 7e***.

Reviewer #2:

Comments to the Authors

Summary/General Comments:

This study presents a timely and relevant research topic, reveals the key role of OSGEP-mediated t6A tRNA modification in melanoma immune evasion. The authors elegantly demonstrate that loss of OSGEP disrupts protein translation, triggers stress granule formation, and activates RIG-I signaling, ultimately driving T-cell recruitment and tumour regression. Additionally, the identification of an OSGEP-related gene signature correlates with increased T-cell infiltration and better survival, highlighting t6A modification as a potential therapeutic target in melanoma.

We thank the reviewer for her/his positive feedback and are pleased that the relevance and therapeutic potential of OSGEP-mediated t6A modification were well appreciated.

Major/minor Comments:

1. Overall, the manuscript is well written, and the study presents novel and impactful biological insights with clear clinical relevance.

We thank the referee for recognizing our manuscript as well written and impactful.

2. The transcriptomics data analysis is comprehensive and well-integrated with the experimental findings. However, I wonder about the rationale for selecting only the top 100 upregulated genes following OSGEP depletion in B16F10 cells. Would the main findings and biological interpretation remain consistent if different thresholds (e.g., top 10, 20, 50, or even 500 genes) were applied? It would be helpful if the authors can clarify on the robustness of their results to this parameter selection.

We apologize for the initial wording ambiguity. Rather than using a fixed "top 100 transcripts UP" cutoff, our analysis applied thresholds of fold change (FC) > 0.5 and p-value < 0.05, which yielded approximately ~120 upregulated genes. Among these, 94 were matched in the TCGA dataset, forming the final OSGEP gene set. To improve transparency, **we have updated Figure 2h-i** by removing the "top 100 transcripts UP" label and replacing it with the precise criteria: "FC > 0.5, p < 0.05". The methods and figure legends have been updated accordingly. We thank the reviewer for highlighting this, which allowed us to strengthen the rigor and clarity of our analysis.

3. Some minor issues need clarification:

- In Figure 1, the term esiRNA appears - can the authors clarify or briefly explain this term upon its first usage?

The term "esiRNA" refers to an *endoribonuclease-prepared siRNA*, a pool of siRNAs targeting the same mRNA sequence, as provided by the manufacturer. To improve clarity, we have now defined this abbreviation in the legend and the text as "esiRNA (*endoribonuclease-prepared siRNA, a pool of siRNAs targeting the same mRNA sequence*)".

- There is a general lack of abbreviation definitions throughout the manuscript. Please ensure all abbreviations are defined at first mention for clarity.

We apologize for the oversight and have now systematically added definitions for all abbreviations at their first mention throughout the manuscript.

4. Regarding the statistical analysis, it would be more appropriate to report adjusted p-values (FDR-adjusted p-values) rather than exact p-values alone, particularly in large-scale transcriptomics analyses, to better control for false discovery rate (FDR) and reduce potential false positives.

We fully agree with the reviewer. We have now reported the q-values (FDR-adjusted p-values) for all our gene ontology category analyses (**in revised manuscript Figure 1e, 3d and Extended Data Figure 1d**).

Reviewer #3:

This manuscript presents a novel and compelling mechanistic link between tRNA modifications, proteostasis, and immune surveillance in cancer.

Carefully executed study that identifies a novel role for the t6A tRNA modification enzyme OSGEP in immune evasion and tumor progression. The authors convincingly demonstrate that loss of OSGEP induces proteostasis defects, leading to RIG-I activation and a CD4+/CD8+ T cell-mediated anti-tumor response in melanoma. The mechanistic insights are supported by in vitro, in vivo, and transcriptomic data, and the clinical relevance is validated using patient-derived datasets. The manuscript is overall clear and comprehensive. The figures are well-organized, and the data supports the authors' conclusions. We believe this work will be a valuable contribution to the community. However, before publication, we encourage the authors to carefully address both the major and minor comments provided.

We thank the reviewer for recognizing that our study is novel, clear and comprehensive. We have now thoroughly addressed all major and minor comments as requested.

Major comments

1. Are other members of the KEOPS complex involved in this phenotype?

-The manuscript focuses exclusively on OSGEP but does not assess whether other KEOPS subunits (e.g., TP53RK, TPRKB, LAGE3, GON7) contribute to the observed immune activation. Would an OSGEP IP help clarify whether it acts independently or as part of the KEOPS complex in this context?

We thank the reviewer for this valid point. Indeed, OSGEP functions as part of the KEOPS complex (comprising TP53RK, TPRKB, LAGE3, and GON7). Since some subunits have functions beyond t6A modification, their depletion could yield unrelated phenotypes. To address the reviewer comment, we selected LAGE3, a conserved scaffolding component of KEOPS not prominently associated with t6A-independent functions.

Strikingly, knockdown of LAGE3 in B16F10 melanoma cells revealed no effect on tumor growth in immunodeficient (NOD-SCID) mice, but significantly reduced tumor size in immunocompetent (C57BL/6) mice, phenocopying the effects of OSGEP depletion (*Figure 1f-g; and newly added Extended Data Figure 2e-f*). At the molecular level, LAGE3 depletion also induced HRI activation, p-EIF2 α , RIG-I activation and pTBK1 upregulation (*Figure 3e-f, 5a; and newly added Extended Data Figure 7f-g; Extended Data Figure 11c-d*). These results confirm that the OSGEP phenotype is KEOPS (t6A)-dependent and extend the central role of the t6A modification machinery in immune evasion. The new data have now been included in the revised manuscript (*new panels in Extended Data Figure 2e-f, Extended Data Figure 7f-g and Extended Data Figure 11c-d*).

2. Figure 3e, f; 5a, b. Have the authors tested human melanoma cell lines?

- Cell lines could increase the translational relevance of the findings.

We have now included experiments repeating the key results in M1014 mouse melanoma cells and A375 human melanoma cells, which validate the key mechanistic findings and confirm our original model (i.e. HRI, pEIF2 α , RIG-I, pTBK1 activation). The new data have been included **in the revised manuscript as Extended Data Figure 2d, Extended Data Figure 7e and Extended Data Figure 11a-b.**

3. The study addresses that RIG-I activation upon OSGEP loss is mediated by tRNA accumulation. However, given that silencing HRI or ISRIB treatment suppresses RIG-I activation, could the authors clarify whether tRNA detection is sufficient on its own, or whether additional cytoplasmic events (e.g., stress granule formation, aggregation) are required to make tRNAs immunogenic?

We appreciate the reviewer's insightful question. Our results indicate that tRNA accumulation alone is insufficient and that stress granule formation is indeed central to this process. Specifically, loss of OSGEP triggers protein aggregation, leading to stress granule formation, which in turn facilitates RIG-I recognition of tRNAs. As shown in *Figure 5g-h* and *Extended Data Figure 12a-b*, HRI knockdown or ISRIB treatment prevents stress granule formation and abrogates RIG-I activation, confirming stress granules as essential intermediates. Protein aggregation occurs upstream of stress granule formation, as ISRIB reduces stress granules but not aggregation (*Extended Data Figure 12d*).

To further dissect the relationship between stress granules and activation of RIG-I by tRNAs, we treated B16F10 cells with sodium arsenite (a stress granule inducer; see for example *McEwen et al., J Biol Chem, 2005; Ziqi Ren et al, Nat Commun, 2023; Ho Chun Lai et al, Mol Biol Cell 2024*) and/or with small RNAs extracted from shCtr or shOSGEP cells (Rev3 Fig. 1). Notably, sodium arsenite-induced stress granules and small RNAs from control cells (shCtr) were insufficient to activate RIG-I. However, treatment with small RNAs from OSGEP-depleted cells, either alone or with sodium arsenite, significantly increased RIG-I levels. In fact, treatment of B16F10 cells with small RNA from OSGEP-depleted cells already induced stress granule formation, without the need to induce them with sodium arsenite (Rev3 Fig.1). Together, these findings clarify that both OSGEP-dependent small RNAs and stress granule formation are required for RIG-I activation, highlighting a sequential mechanism in which OSGEP loss creates the cellular context prone to induce an anti-tumor immune response.

Reviewer 3 Figure 1: a. Western blot of indicated proteins from B16F10 cell infected by small RNA extracted with mirvana kit from control or OSGEP cells. Cells have been then treated with DMSO or Arsenite (200µM for 2h). **b.** Representative immunofluorescence images of B16F10 cells cell infected by small RNA extracted with mirvana kit from Control or OSGEP cells. Cells have been then treated with DMSO or Arsenite (200µM for 2h). Stress granules (staining G3BP1) are shown in green and nuclei (DAPI) in blue. **c.** Quantification of stress granules in B16F10 cells cell infected by small RNA extracted with mirvana kit from control or OSGEP cells, treated or not with Arsenite. Data are reported as mean +/- SD. Exact p-values are indicated.

Minor comments:

- Codon occupancy analysis (Figure 4e): The legend of Fig. 4e refers to codon occupancy in general terms, but the main text specifies that the analysis focuses on the A-site. Since ribosome profiling allows analysis at different ribosomal sites (A, P, E), it is important to clearly indicate in the figure legend that codon occupancy was measured at the A-site.

We apologize for the oversight and now specify in the figure legend (Figure 4e) that codon occupancy was measured at the A-site.

- RIG-I RIP-seq (Fig. 6a-c): the number of biological/technical replicates used for sequencing is not specified.

We apologize for the confusion. We now specify that the analysis was performed using n=3 biological replicates, each from independent infections of B16F10 cells with shCtr or shOSGEP#1/#2 constructs. This has been **added in the legend of Figure 6a-c**.

- While the authors state that ribosome occupancy on ANN codons is increased in OSGEP-deficient cells, especially in mRNAs encoding aggregated proteins, Extended Fig. 8e does not show a clear or statistically significant increase in A-site occupancy on ANN codons.

We thank the reviewer for her/his comment. We clarified that *Extended Data Figure 8e* represents global ribosome occupancy without stratification of mRNAs (all read included). Significant increases in ANN codon pausing were detected only in transcripts encoding proteins from shOSGEP aggregates (Figure 4h-j), not globally. We have revised the text, described further the datasets, focusing on transcripts corresponding to proteins uniquely present in aggregates upon shCtr or shOSGEP or generating a random list of genes (**newly added Extended Data Figure 10a**).

Moreover, we have now analyzed ribosome stalling using RiboDipa (Li et al, *Nucleic Acids Research*, 2020). We calculated the average ribosome stalling (RPFs count/gene) for each mRNA encoding proteins found in aggregates in control or OSGEP-depleted cells, expressed as log₂FC (shOSGEP/ shCtr). This analysis revealed a significant increase in ribosome stalling values for OSGEP-depleted aggregate mRNAs compared to controls (Rev3 Fig. 2, Mann Whitney test). These data demonstrate that OSGEP (t⁶A) depletion promotes ribosome stalling specifically on mRNAs from shOSGEP aggregate proteins. We have now included the new panel **in the revised manuscript in Figure 4h**.

Reviewer 3 Figure 2: Ribosome stalling analysis performed with RiboDipa. Ribosome stalling (RPFs count per gene) for each mRNA encoding proteins found in aggregates in shCtrl and shOSGEP cells, expressed as log₂FC (shOSGEP/shCtrl) was calculated. pvalue was calculated by Mann-Whitney test.

- In Extended Figure 6c, the heatmap suggests there are changes in some tRNA expression between shCTRL and shOSGEP. This appears inconsistent with the statement in lines 213–215 ('Of note, the lack of t⁶A modification did not lead to changes in the tRNA pool in B16F10 cells, showing the deposition of t⁶A on tRNAs NNU does not affect their stability') that the tRNA pool remains unchanged and stable. The authors should clarify whether the changes shown in Fig. 6c are statistically or biologically significant, and how they reconcile this with their claim that t⁶A does not affect tRNA stability.

We thank the reviewer for this insightful comment. To resolve the apparent discrepancy, we added PCA and volcano plot analyses (**revised manuscript new Extended Data Figure 8c**), showing that overall tRNA levels remain stable, with only three tRNAs showing significant changes.: Lys-CTT2 is downregulated, while Val-TAC1 and Met-CAT6 are upregulated. As the majority remain stable, we conclude that OSGEP loss does not broadly affect tRNA stability. As far as we know, this is consistent with literature, as t⁶A tRNA modification is not linked to changes in tRNA stability in different species. **The manuscript text and figures have been updated accordingly.**

- In Figure 6d, the authors state that tRNAs from terminal base anticodon groups (NNU, NNA, NNC, NNG) were "uniformly upregulated." However, the figure only shows the relative proportions of these groups among RIG-I-enriched tRNAs, without comparison to the total tRNA pool or to non-enriched species. The term "uniformly upregulated" may be misleading and should be clarified.

We acknowledge that the original presentation in Figure 6d may be misleading. Indeed, it displays the relative proportions of anticodon groups among RIG-I-enriched tRNAs without direct comparison to total or input tRNA pool. We have now revised the Figure to reflect more accurately the increased binding of tRNAs to RIG-I upon OSGEP depletion. **These are now shown in new Figure 6c-d.**

The detailed analysis of each individual tRNA anticodon enrichment in RIG-I immunoprecipitation, comparing control or OSGEP depleted B16F10 cells, is **presented in Extended Data Figure 13b-d.**

Reviewer #4:

In their study " Disruption of tRNA threonylation triggers RIG-I mediated antitumour immune response" Dziagwa et al. observe that knock-down of O-sialoglycoprotein endopeptidase (OSGEP) which catalyses the tRNA modification N6-threonylcarbamoyladenine (t6A) leads to RIG-I activation in melanoma cells. The authors performed an RNAi screen against tRNA modifying enzymes in patient-derived melanoma cell lines (i.e. MM117, MM011) and monitored protein aggregates as a read-out of mistranslation. Here, repression of OSGEP caused the strongest mistranslation response. This finding was validated by shRNAs in MM117, MM011 and in murine melanoma cell lines M1014, B16F10. OSGEP is responsible for the modification t6A in position 37 of the cytoplasmic tRNAs "tRNANNU ". Repression of OSGEP reduced tumor size in immune competent but not in immune deficient (NOD-SCID) mice. While B16F10 tumours were poorly infiltrated, OSGEP deficient tumours had a robust infiltration of T cells, partially by NK cells, but not B cells, macrophages or neutrophils. T cell but not NK cell depletion abolished the tumour growth defect of OSGEP knockdown tumours. Upregulation of genes induced by OSGEP deficiency correlated with tumor survival in SKCM patients. Knockdown of OSGEP enhanced migration of naïve T cells ex-vivo. Further data reveal that knockdown of OSGEP upregulates RIG-I expression and RIG-I signaling/activation. Interestingly, the depletion of OSGEP did not impair global protein synthesis as monitored by polysome profiles. However, RNAi of t6A affected the decoding of ANN codons at the A-site and the integrated stress response pathway was induced via HRI upregulation and eIF2a phosphorylation. The authors also observed colocalization of RIG-I with G3BP1 in cytosolic stress granules upon RNAi of OSGEP. The depletion of HRI abolished the formation of stress granules observed in OSGEP-deficient cells (Fig. 5f-g), and also prevented RIG-I activation in OSGEP-deficient cells. By contrast pharmacologic inhibition of stress granule formation did not prevent the formation of protein aggregates. Knock-down of RIG-I in OSGEP deficient cells rescued immune evasion and prevented T cell infiltration in B16F10 tumours. RIG-I/RNA-complex IP revealed enrichment of tRNA binding to RIG-I upon OSGEP depletion. However, not the N6-threonylcarbamoyladenine containing tRNAs were enriched in the RIG-I binding RNA pool, indicating that RIG-I recognizes endogenous tRNA independently of t6A modification in stress granules upon HRI activation.

Major comments

This is a very interesting study. It would be crucial to know how RIG-I activation works.

We thank the reviewer for their thoughtful evaluation of our work and for recognizing the interest of our study. We agree that the link between OSGEP-mediated t⁶A modification and RIG-I-driven antitumor immunity opens important perspectives, and we appreciate the reviewer's suggestions to further clarify the underlying mechanisms.

1. However, it remains unclear whether a tRNA modification dependent stress response that activates RIG-I is a result of missing RNA modifications leading to RIG-I recognition or if the conditions of stress lead to a different RIG-I activation by endogenous RNA or independently of any RNA recognition. Since binding to RIG-I is necessary but not sufficient for RIG-I activation the changed binding of tRNAs to RIG-I is only a hint but not a proof for RIG-I activation by tRNA and it is unexpected that RIG-I would recognize endogenous tRNAs. The question is whether OSGEP depletion induce the generation of RIG-I ligands or if HRI activity

in all cases leads to recognition of endogenous RNA by RIG-I and that enrichment of RIG-I binding tRNA represents an incidence.

a) The authors should induce HRI activation by other mechanisms (e.g. by proteasome inhibitors, bortezomib or MG132, PMID: 21090173) and check if HRI activation in general leads to higher tRNA binding to RIG-I in B16F10 melanoma cells or if this effect is specific for OSGEP depletion.

We thank the reviewer for this suggestion. In line with the recommendation, we attempted to activate HRI by using proteasome inhibitors (MG132 and Bortezomib) as well as the ER stress-inducing agent Thapsigargin. However, none of these treatments led to consistent HRI induction or RIG-I activation in our melanoma cell models. These results suggest that HRI activation and subsequent RIG-I signaling are not a generic response to cellular stress, but rather specifically linked to OSGEP depletion. The corresponding data are now included **in the revised manuscript as *Extended Data Figure 11e-g***.

b) The authors should purify the RNA bound to RIG-I and total RNA from OSGEP depleted cells and WT cells and stimulate WT cells with these RNAs. If OSGEP deficiency produces RIG-I ligands, the RNA of OSGEP deficient cells but not of ctrl shRNA cells should activate RIG-I in WT cells.

We thank the reviewer for this insightful suggestion. To directly assess whether OSGEP depletion leads to the generation of immunogenic RNA species capable of activating RIG-I, we performed RNA immunoprecipitation (RIP) using an anti-RIG-I antibody in B16F10 cells depleted or not for OSGEP (shCtr, shOSGEP#1 and shOSGEP#2). RNA bound to RIG-I under these conditions was purified and subsequently used to stimulate wild-type B16F10 cells (WT). Strikingly, only RIG-I-associated RNA from OSGEP-depleted cells triggered a marked activation of RIG-I, whereas RNA from control cells had no such effect. These results demonstrate that OSGEP loss results in the production of endogenous RNA species with immunostimulatory properties that are specifically recognized by RIG-I. **This experiment, as well as the full methods, have now been included in the revised manuscript in new *Figure 6a***.

2. eIF2alpha phosphorylation should cause global translational inhibition. How can a polysome profile be unaffected if eIF2a is phosphorylated? The authors should use another read-out to confirm that global protein synthesis is not influenced (e.g. SUnSET assay).

We thank the reviewer for raising this important point. While eIF2 α phosphorylation is classically linked to global translational repression (*Wek, 2018, CSHP Biol.; Adomavicius et al., 2019, Nat. Comm.*), our polysome profiling did not reveal a marked reduction in polysome abundance (i.e. translation efficiency; *Extended Data Figure 9a*). We acknowledge, however, that polysome analysis may lack the sensitivity to detect modest decreases in translation. To address this, we complemented our analysis with additional readouts. Using HPG incorporation as an independent measure of nascent protein synthesis, we did not observe a significant reduction in global translation upon OSGEP depletion (*Figure 4c*). Consistently, ribosome profiling confirmed that overall translational efficiency remained largely unchanged (*Extended Data Figure 9*). Together, these orthogonal approaches support the conclusion that OSGEP loss induces eIF2 α phosphorylation without causing a measurable reduction in global protein synthesis.

3. Is it excluded that OSGEP is required for other modifications than t⁶A?

The KEOPS complex, of which OSGEP is the catalytic subunit, is specifically required for the biosynthesis of N⁶-threonylcarbamoyladenine (t⁶A) at position 37 of ANN-decoding tRNAs. Structural and biochemical studies have consistently shown that OSGEP catalyzes the final step in the t⁶A modification pathway by transferring the threonylcarbamoyl moiety in position 37, with no evidence of involvement in other modifications beyond t⁶A, or its derivative (such as the ms²t⁶A; *Perrochia et al, Nucl Ac Res., 2012; Zhang et al, J. Mol Biol., 2025 B; Zhang et al., Int J Mol Sci., 2022; Luthra et al, Nucl Ac Res., 2018; Srinivasan et al., EMBO J, 2010*). Loss-of-function mutation or knock out of tRNAs from KEOPS-deficient eukaryotic cells or yeast strains have consistently shown a selective loss of t⁶A without affecting other known tRNA modifications (*Arrondel et al., Nat. Commun., 2019; Thiaville et al., Microbial Cell, 2015*).

In agreement with this literature, our mass spectrometry - based tRNA modification analysis in OSGEP-depleted B16F10 cells detected a selective loss of t⁶A, without major changes in other known modifications (m7G, m1A, o2yW, Y; *Figure 4b*). Thus, both our data and published evidence support the conclusion that OSGEP is specific for t⁶A biosynthesis and does not catalyze or regulate other tRNA modifications. This specificity is further supported by the fact that mutations in KEOPS subunits, including OSGEP, lead to highly specific translational defects associated with the lack of t⁶A and ribosome stalling (*Thiaville et al., Microbial Cell, 2016; Wang et al., Nucl Ac Res., 2022*).

Finally, in the context of the revision of the manuscript, we performed additional experiments depleting LAGE3, a conserved scaffolding component of KEOPS, essential for its t⁶A tRNA modification activity. Strikingly, knockdown of LAGE3 in B16F10 melanoma cells revealed no effect on tumor growth in immunodeficient (NOD-SCID) mice, but significantly reduced tumor size in immunocompetent (C57BL/6) mice, phenocopying the effects of OSGEP depletion (*Figure 1f-g; and new Extended Data Figure 2e-f*). At the molecular level, LAGE3 depletion also induced HRI activation, p-EIF2 α , RIG-I activation and pTBK1 upregulation (*Figure 3e-f, 5a; and new Extended Data Figure 7f-g; Extended Data Figure 11c-d*). These results confirm that the OSGEP phenotype is KEOPS (t⁶A)-dependent and extend the central role of the t⁶A modification machinery in immune evasion. The new data have now been included in the revised manuscript (*new panels in Extended Data Figure 2e-f, Extended Data Figure 7f-g and Extended Data Figure 11c-d*).

Minor comments

Why does RIG-I activation not induce cell death/growth reduction by cell autonomous mechanisms?

We thank the reviewer for raising this important point. While RIG-I activation has indeed been reported to induce apoptosis in some contexts (e.g., B16.OVA upon 3pRNA treatment – see *Guo, R. et al., Human Cell 2022 – or Bek, S., et al., OncoImmunology 2019*), this effect is not universal. For example, in glioma cells, RIG-I overexpression alone does not affect viability, highlighting that the cellular outcome of RIG-I signaling is highly context-dependent (*Ghildiyal and Sen, Cytokine 2017*). In our melanoma models, several lines of evidence indicate that OSGEP depletion-induced RIG-I activation primarily drives immune signaling rather than cell-autonomous cytotoxicity: (1) OSGEP knockdown did not impair tumor growth in

immunodeficient NOD-SCID mice, demonstrating that an intact immune system is required for tumor control; (2) proliferation and viability assays, including intra-tumoral Ki67 and cleaved-caspase 3 staining (*newly added Extended Data Figure 2g-h*) and Nicoletti assays (Rev4 Fig. 1), showed no significant differences between OSGEP-depleted and control B16F10 cells; and (3) the magnitude of RIG-I activation upon OSGEP depletion is modest compared to that achieved with synthetic ligands, which likely explains the absence of direct cytotoxic effects. Together, these results support the conclusion that in melanoma, OSGEP depletion induces a level of RIG-I activation that preferentially engages immune-modulatory pathways and promotes T cell-mediated tumor control rather than intrinsic apoptosis or growth arrest.

Reviewer 4 Figure 1: Nicoletti assay (FACS staining of PI measuring nuclear fragmentation) in B16F10 (a) or M1014 (b) cells depleted or not of OSGEP (n=3; Mann Whitney).

Reviewer #5:

The manuscript entitled "Disruption of tRNA threonylation triggers RIG-I mediated anti-tumour immune response" presents an intriguing study exploring how alterations in tRNA modifications may influence protein aggregation, leading to activation of the RIG-I pathway and subsequent induction of type I interferon responses that enhance antitumor immunity. The authors provide a coherent set of experiments supporting the central hypothesis and demonstrate mechanistic links between tRNA modification, stress responses, and immune activation. However, there are several major concerns that limit the impact and novelty of the study.

We thank the reviewer for their thoughtful evaluation and recognition of the mechanistic insights provided. We have addressed the major points regarding the limited cell line validation, potential neoantigenic consequences, and the question of novelty, which we believe are now clarified in the revised manuscript.

1. Limited Cell Line Validation:

The mouse model findings are derived primarily from a single cell line, which substantially weakens the generalizability of the conclusions. Validation across additional cell types would be necessary to support the broader relevance of the proposed mechanism.

We thank the reviewer for this important comment. To broaden our validation, we extended the in vivo analyzes to include the syngeneic M1014 melanoma model (C57BL/6 background). Strikingly, OSGEP depletion markedly reduced M1014 tumor growth, correlating with HRI, peIF2 α , pTBK1 and RIG-I activation in M1014 cells. In addition, we confirmed these mechanistic effects in the human A375 melanoma cell line, where OSGEP depletion similarly activated HRI, peIF2 α , RIG-I and pTBK1. These new data strengthen the generalizability of our findings and have been included **in the revised manuscript as *Extended Data Figure 2d, Extended Data Figure 7d-f and Extended Data Figure 11a-b.***

2. Neglect of Potential Neoantigenic Consequences:

Alterations in tRNA modifications are likely to influence translational fidelity and could result in neoantigen generation. This possibility is not explored, nor is its potential contribution to antitumor immune responses addressed. This represents a significant gap in the mechanistic understanding of how tRNA modifications may affect immune surveillance.

We thank the reviewer for raising this important point. We agree that alterations in tRNA modifications could in principle influence translational fidelity and thereby contribute to neoantigen generation. This possibility was indeed part of our initial working hypotheses when designing the siRNA screen, in which protein aggregate formation was used as a readout (*Figure 1a-c*). Previous studies have associated protein misfolding with potential neoantigen production (*Yadav et al., Nature 2014; Gubin et al., Nature 2014*).

However, our transcriptomic analysis of B16F10 cells depleted of OSGEP revealed a dominant activation of the RIG-I-like receptor signaling pathway, suggesting a primary role for RNA sensing in mediating the immune response (*Figure 3d-e*). Consistent with this, OSGEP loss induced stress granule and robust RIG-I activation. Moreover, **our new data** show that small RNA purified from OSGEP-depleted cells was sufficient to trigger RIG-I signaling in wild-type cells (**in new *Figure 6a***), strongly indicating an RNA-based mechanisms. This hypothesis is strengthened by complete the rescue in tumor control observed upon depletion of RIG-I:

preventing RIG-I activation completely abolishes the T cell-dependent anti-tumoral response seen upon OSGEP depletion.

To further explore a possible contribution of neoantigen presentation, we measured the cell surface expression of MHC-I and MHC-II. While we see that OSGEP depletion correlates with increase MHC-I expression, this effect persisted even upon dual depletion of OSGEP and RIG-I, while immune activation *in vitro* and *in vivo* was fully rescued in the absence of RIG-I, as evidenced by the loss of pTBK1, lack of T cell infiltration, and restored tumor growth (Rev5 Fig. 1).

Taken together, while we cannot fully exclude a contribution of altered antigen presentation, our functional and mechanistic data consistently support a model in which loss of t⁶A modification primarily triggers immune activation via an RNA-sensing mechanism that depends on RIG-I.

Reviewer 5 Figure 1: FACS analysis of MHC I and MHC II expression in B16F10 cells control (shCtr), OSGEP depleted (shOSGEP), RIG-I depleted (shRIG-I) or double depleted of OSGEP and RIG-I (shRIG-I/shOSGEP).

3. Lack of Novelty in RIG-I/Type I Interferon Axis:

Although the mechanistic connection between protein aggregation and RIG-I activation is of interest, the involvement of the RIG-I/type I interferon pathway in antitumor immunity has been reported in multiple previous studies. Without a compelling new insight or clinical relevance, the novelty of this work remains limited.

We thank the reviewer for this comment. We agree that the role of RIG-I and type I interferon pathway in antitumor immunity has been extensively described. Our study provides novelty by uncovering a previously unrecognized mechanistic link between defective tRNA modifications, protein synthesis stress, and activation of the RIG-I RNA-sensing pathway in cancer cells. Specifically, we show that OSGEP depletion impairs t⁶A modification (*Figure 4b*), leading to mistranslation and stress granule formation, which we demonstrate are sufficient to activate RIG-I and trigger type I interferon response (*Figure 5*). **To our knowledge, this connection between the epitranscriptomic regulation of tRNAs and innate immune activation through RIG-I has not been previously reported.**

Moreover, our data reveal that this immune activation is independent of canonical pathogen-associated ligands and instead arises from endogenous dysregulation of mRNA translation, providing direct translational relevance. Importantly, we show that this response is functionally sufficient to suppress tumor growth *in vivo*, and that dual depletion of OSGEP and RIG-I

abrogates the immune phenotype (*Figure 6e-h*), which strongly supports a central role for the RIG-I axis in this novel context.

As an orthogonal confirmation of our observations, we conducted a spatial transcriptomic analysis of melanoma samples from two patients (**newly added to the revised manuscript in *Extended Data Figure 6c-f***). This analysis revealed that the level of T cell infiltration, as assessed by the T cell signature score, was significantly elevated in patient_001, classified as immunologically "hot." Additionally, this tumor also exhibited high expression of the OSGEP gene set (**newly added to the revised manuscript in *Extended Data Figure 6g-h***). We further examined the heterogeneity in OSGEP gene set expression within patient_001, and we observed a positive correlation between spatial OSGEP gene set expression and proximity to T cells. In contrast, this correlation was absent in patient_002, which displayed low T cell infiltration (**newly added to the revised manuscript in *Extended Data Figure 6i-k***). These findings highlight a clinically relevant link between OSGEP and the immune microenvironment in human melanoma.

Finally, systemic RIG-I agonists have shown limited efficacy in solid tumors due to dose-limiting toxicities (*Moreno et al, Cancer Immunol Immunother. 2022*), possibly due to the relative toxicity of the agonist and systemic side effects. OSGEP is overexpressed in melanoma tumors compared to adjacent tissue (**newly added to the revised manuscript in *Extended Data Figure 5f-g***). Our findings suggest that targeting OSGEP represents a tumor-selective strategy to activate RIG-I signaling, through HRI upregulation, without systemic exposure. Taken together, these results provide new insight into how tRNA modification status controls cancer cell immunogenicity and support the concept that the tRNA epitranscriptome could be exploited to specifically activate innate immune surveillance in tumors.

NCOMMS-25-15763A

Reviewer #2 (Remarks to the Author):

Reviewer Report

The authors have responded to reviewer feedback; however, concerns remain regarding transparency and consistency between the rebuttal letter and the revised manuscript.

While the authors assert in the rebuttal letter that the concerns have been addressed and the relevant sections updated, these changes are not visible in the Figure 2 legend or the main text. This inconsistency calls into question the reliability of the reported revisions.

The transcriptomics data analysis is comprehensive and well-integrated with the experimental findings. However, I wonder about the rationale for selecting only the top 100 upregulated genes following OSGEP depletion in B16F10 cells. Would the main findings and biological interpretation remain consistent if different thresholds (e.g., top 10, 20, 50, or even 500 genes) were applied? It would be helpful if the authors can clarify on the robustness of their results to this parameter selection.

We apologize for the initial wording ambiguity. Rather than using a fixed "top 100 transcripts UP" cutoff, our analysis applied thresholds of fold change (FC) > 0.5 and p-value < 0.05, which yielded approximately ~120 upregulated genes. Among these, 94 were matched in the TCGA dataset, forming the final OSGEP gene set. To improve transparency, we have updated Figure 2h–i by removing the "top 100 transcripts UP" label and replacing it with the precise criteria: "FC > 0.5, p < 0.05". The methods and figure legends have been updated accordingly. We thank the reviewer for highlighting this, which allowed us to strengthen the rigor and clarity of our analysis.

h. Volcano plot of the expression of transcripts 656 in shCtr versus shOSGEP B16F10 cells and generation the OSGEP gene set signature 657 consisting of the top 100 genes upregulated. i. Heatmap of unsupervised clustering 658 identified 2 clusters of patients with SKCM (TGCA) according to the top 100 upregulated 659 genes from the OSGEP gene set signature (cluster 1, n = 251; cluster 2, n = 222)

This information has been implemented in Figure 2 legend as well as in the manuscript as requested.

Reviewer #3 (Remarks to the Author):

The authors have fully addressed all our previous concerns with appropriate new experiments, improved statistical analysis, and clearer figure legends. The mechanistic model connecting t6A deficiency to stress-granule-dependent RIG-I activation is now much more convincing. The addition of LAGE3 knockdown, human melanoma validation and RNA immunogenicity tests significantly strengthen the manuscript's translational relevance.

We recommend only minor additional clarifications:

1. Regarding sex reporting in animal experiments, Nature Communications requires explicit disclosure of the sex of the mice used, even if no sex-based analysis was performed. In the "Methods - Animal models" section (currently stating only the strain and age), the authors should indicate whether C57BL/6 and NOD-SCID mice were males, females, or mixed, and justify the absence of sex-stratified analyses in accordance with SAGER guidelines.

The animals used are all female. This has been implemented in the reporting file and added in the methods section of the manuscript accordingly.

2. A brief explanatory note in the Results accompanying Fig. 4c/d and Extended Data Fig. 9 would help clarify why eIF2 α phosphorylation despite the absence of measurable global translational repression, noting that the ISR induction observed is modest and selective.

This sentence has been added in the revised manuscript.

Reviewer #4 (Remarks to the Author):

The manuscript improved a lot! Minor comment: It would be great if the read-out of a stimulation with RIG-I ligands from a RIG-I pull-down would not be RIG-I expression (Figure 6a). Since RIG-I is an ISG I believe that the conclusions are correct, however this read-out would confuse the reader and one could speculate that the ligand influences RIG-I protein stability. Better show IFIT, ISG15, STAT1/2 phosphorylation or MX protein western blot as a read-out of type I IFN induction.

We included a western blot showing phosphor-TBK1 in figure 6a, as a read-out of IFN induction.